# TOR regulates variability of protein synthesis rates

Clovis Basier [ID][1]✉ & Paul Nurse[1,2]

## Abstract

**Cellular processes are subject to inherent variability, but the extent to which cells can regulate this variability has received little investigation. Here, we explore the characteristics of the rate of cellular protein synthesis in single cells of the eukaryote fission yeast. Strikingly, this rate is highly variable despite protein synthesis being dependent on hundreds of reactions which might be expected to average out at the overall cellular level. The rate is variable over short time scales, and exhibits homoeostatic behaviour at the population level. Cells can regulate the level of variability through processes involving the TOR pathway, suggesting there is an optimal level of variability conferring a selective advantage. While this could be an example of bet-hedging, but we propose an alternative explanation: regulated 'loose' control of complex processes of overall cellular metabolism such as protein synthesis, may lead to this variability. This could ensure cells are fluid in control and agile in response to changing conditions, and may constitute a novel organisational principle of complex metabolic cellular systems.**

**Keywords** Variability; Protein Synthesis; Single-Cell; TOR Pathway; Control
**Subject Categories** Cell Cycle; Metabolism; Translation & Protein Quality

## Introduction

Isogenic populations of bacterial to metazoan cells growing in constant conditions exhibit cell-to-cell phenotypic heterogeneity: for example, resistance to antibiotics in bacteria (Davidson and Surette, 2008), to antifungals in yeasts (Bojsen et al, 2017), and to drug treatments in cancer cells (Inde and Dixon, 2018; Vallette et al, 2019). Such variability is generally explained in terms of bet-hedging strategies (Philippi and Seger, 1989), whereby some traits that confer a fitness burden during optimal growth conditions provide an advantage under changing conditions. This has usually been applied to relatively simple metabolic processes consisting of a few reactions such as the synthesis of proteins from single genes, and is often explained by stochastic gene expression (Eldar and Elowitz, 2010). In addition, metabolic variability, including the overall growth of cells and ATP concentration, has been suggested

as another source of phenotypic variability in bacteria (Kiviet et al, 2014; Lin and Jacobs-Wagner, 2022), although variability in ATP levels was not observed in eukaryotic yeast cells (Takaine et al, 2019). Here we have investigated in the single-cell eukaryote fission yeast *Schizosaccharomyces pombe*, the dynamics of the variability of the net measured rate of protein synthesis, which extends from the rate of uptake of amino acids to their incorporation into nascent proteins and subsequent protein turnover. Exogenous labelled amino acids have long been used to measure the rates of protein synthesis in populations of cells (Creanor et al, 1982; Fan and Penman, 1970), and click chemistry technical advances now allow the single-cell measurement of incorporation of amino acid into total cellular protein (Kolb et al, 2001; Beatty et al, 2006; Basier and Nurse, 2023). To explore the variability and organisational characteristics of protein synthesis in eukaryotes, a process which involves expression of many hundreds of genes and individual reactions, we have made single-cell measurements of the flux of exogenous amino acids into nascent proteins, and have used this to investigate the variability in the rate of protein synthesis and discuss the implications of this variability for the regulation of global cellular metabolic processes.

## Results

### Cell-to-cell variability in steady-state growing cells

A single-cell fluorescence assay was used to measure the rate of total cellular protein synthesis of steady-state cultures of fission yeast cells based on the uptake and incorporation of an exogenous methionine analogue into newly synthesised proteins (Basier and Nurse, 2023) (Fig. 1A). Exponentially growing populations of cells were incubated for 5 min with Homopropargylglycine (HPG), a 'clickable' methionine analogue (Beatty et al, 2006), then fixed with formaldehyde, and the HPG molecules incorporated into nascent peptides were fluorescently labelled (Fig. 1B). Bright-field and fluorescence microscopy were used to obtain single-cell measurements of cell size and HPG incorporation. A surprisingly high cell-to-cell heterogeneity in HPG signal was observed within the population (Fig. 1C,D). For all cell sizes, the HPG signal in different cells varied more than threefold with a long-tailed distribution (Fig. 1E), resulting in a coefficient of variation (CV) of over 30% (Fig. 1F). This heterogeneity was observed at all cell cycle stages within asynchronous populations (Basier and Nurse, 2023) (Fig. EV1A–D). Using two different methionine analogues, we showed that the staining procedure was not responsible for the long

[1]Cell Cycle Laboratory, The Francis Crick Institute, London NW1 1AT, UK. [2]Laboratory of Yeast Genetics and Cell Biology, Rockefeller University, New York, NY 10065, USA.
✉E-mail: clovis.basier@crick.ac.uk

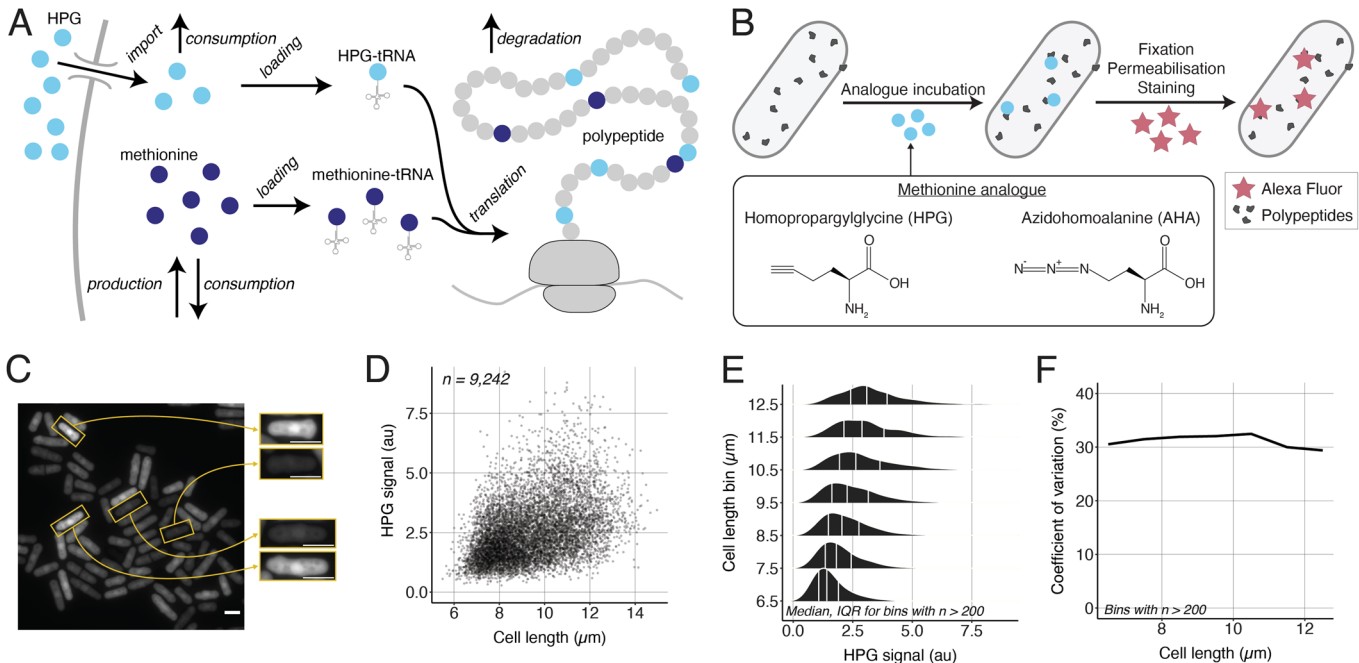

**Figure 1.  Cell-to-cell variability of the measured global protein synthesis rate in exponentially growing unperturbed fission yeast populations.**

(A) Overview of the metabolic process assayed, from the HPG uptake through to its incorporation into nascent polypeptides. (B) Schematic of the assay used. (C) Example of fluorescence images of exponentially growing wild-type fission yeast (PN1) cells assayed. Cells of approximately the same length but strikingly different HPG signals are featured on the right. The scale bars represent 5 μm. (D) Single-cell HPG signal in an exponentially growing wild-type fission yeast (PN1) population of cells. (E) Cells shown in (D) are grouped in bins of 1 μm (centre value for the bin shown) and the kernel density estimate of the single-cell HPG signal is shown for each bin. The white lines show the first, second, and third quartiles of the population. (F) CV of the HPG signal for the different length bins shown in (E). Source data are available online for this figure.

tail in the distribution (Fig. EV1E,F). We conclude that this high cell-to-cell heterogeneity in measured protein synthesis is not restricted to a particular cell size, or cell cycle stage of a cell.

## Single-cell dynamics of variability

Next, we investigated if the population heterogeneity reflected transient or long-lasting differences between cells. We used pulses of HPG and Azidohomoalanine (AHA), another 'clickable' methionine analogue (Kiick et al, 2002) (structure in Fig. 1B) which can be stained independently of HPG, to determine the rate of measured protein synthesis at two different times within the same cell. Thus, the HPG signal represents the rate of measured protein synthesis at the beginning of the experiment, whilst the AHA signal represents the rate at a later time (Fig. 2A). We found that the mean HPG and AHA signals per cell were well correlated when both analogues were pulsed at the same time ($R^2 = 0.85$), but rapidly became un-correlated as the time between the addition of the two analogues increased, with the $R^2$ between the two signals dropping to 0.46 when AHA was added 20 min after HPG and further dropping to 0.12 when the interval was 50 min (Fig. 2B–D). This rapid de-correlation of the signals was clearly visible in the fluorescence images (Fig. 2C). These results indicate that the rate of the measured global protein synthesis is dynamic, with a relatively short time scale of around 20 min. The mean AHA signal of cells ranked and grouped based on their HPG signal regressed towards the mean of the whole population within 40–60 min (0.15–0.25 of

the generation time). This was the case for groups of cells with both high and low rates of the measured protein synthesis at the beginning of the experiment (Figs. 2E–G and EV2A,B). These results indicate that cells with a measured global protein synthesis rate that is distant from the mean of the population, tend to adjust their synthesis rate to approach the mean population rate over a relatively short time period, reflecting a homoeostatic behaviour at the population level. In addition, the differences in correlation between the HPG and the AHA signal with different timing intervals, indicate that the observed cell-to-cell variability does not arise from differences in cell permeabilisation before the staining procedure, indicating that the observed variability reflects physiological variability within the living cells rather than variability in our measurement technique.

Given these results, we predicted that the variability in the population should reduce if cells are exposed to HPG for a long period of time. To test this, we added HPG 10 separate times over different lengths of times ranging from 10 to 180 min and looked at the dispersion of the distribution of the single-cell HPG signals at the end of the incubation period (Fig. EV2C–F). We expressed the level of variability in the population as the quartile coefficient of dispersion (QCD) (Bonett, 2006) of the single-cell HPG signal distribution. QCD is an analytical procedure which avoids emphasis on the contributions of extreme outlier cells to the measurement of variability. We found that the QCD of the HPG signals within the population decreased as the length of incubation increased (Fig. EV2G), supporting the idea that cells with a high or

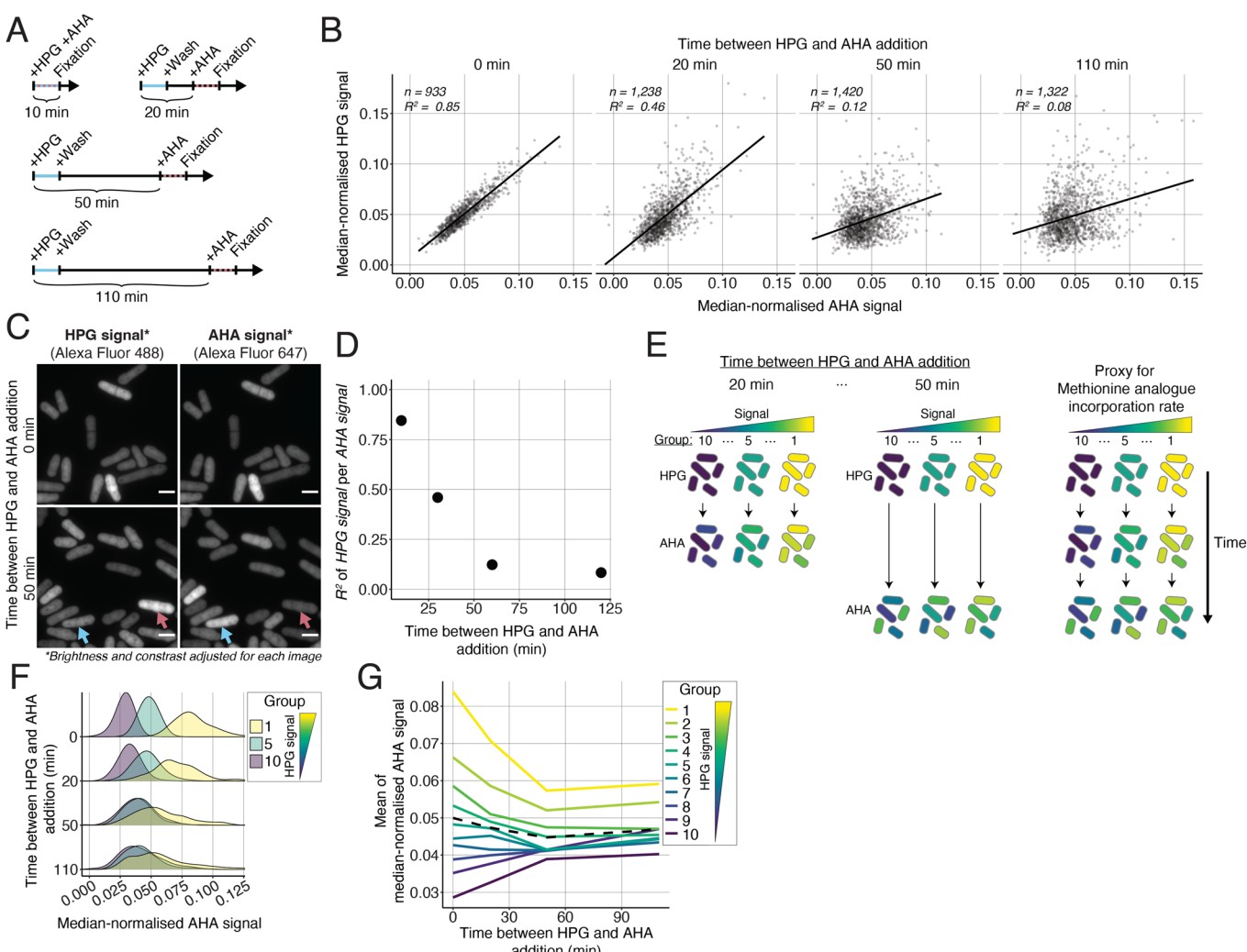

**Figure 2.  Dynamics of single-cell variability in the measured global protein synthesis rate.**

(A) Schematic of the HPG and AHA double pulsing experiments. (B) For each experiment shown in (A), the single-cell HPG and AHA signals are divided by the cell length and normalised to the median HPG and AHA signals of the population (black dots). The black line is the ordinary least square (OLS) linear regression fitted on the single-cell data. (C) Example images of HPG and AHA labelling signals. The red and blue arrow show cells with strikingly different signal in the HPG and the AHA channels when AHA is added 50 min after HPG. The scale bars represent 5 µm. (D) The coefficient of determination ($R^2$) of the OLS linear regressions shown in (B) as a function of the time between HPG and AHA addition. (E) Schematic of the data processing approached used for (F, G). (F) Kernel density estimates of the median-normalised AHA signal divided by cell length (x axis in (B)) of the groups computed using the approach shown in (E). Only the 1st, 5th, and 10th groups are shown (see Fig. EV2B for all groups). (G) For each group computed using the approach shown in (E), the mean of the median-normalised AHA signal divided by cell length is calculated and shown as a function of the time between HPG and AHA addition. The dashed line represents the mean trajectory of the overall population. Source data are available online for this figure.

low HPG signal are in a transient state, and that over longer time scales the measured global protein synthesis of a cell approaches the mean of the population.

## Genetic determinant of variability

We next considered whether this variability is an inherent unchangeable consequence of the random nature of biochemical reactions, or if it was subject to active cellular regulation. We hypothesised that if cells regulated the variability in their metabolism, there would be cellular pathways involved in controlling the level of variability. Thus, mutations in genes involved in these pathways would result in higher or lower

population heterogeneity in the measured rates of protein synthesis. To increase the likelihood of finding such mutations, we included gene mutations in the TOR pathway and cell cycle control since the TOR pathway has been implicated in a wide range of metabolic and growth control processes in eukaryotic cells (Dibble and Manning, 2013; Wullschleger et al, 2006), and we have previously shown that the measured rate of protein synthesis is subject to some regulation at different phases of the cell cycle (Basier and Nurse, 2023). We also included strains harbouring mutations in genes related to other biological processes. To minimise effects coming from changes in cell physiology and to allow a more direct comparison with the wild-type strain, we assayed only prototrophic strains with no apparent growth defects. All strains were cultured in the same

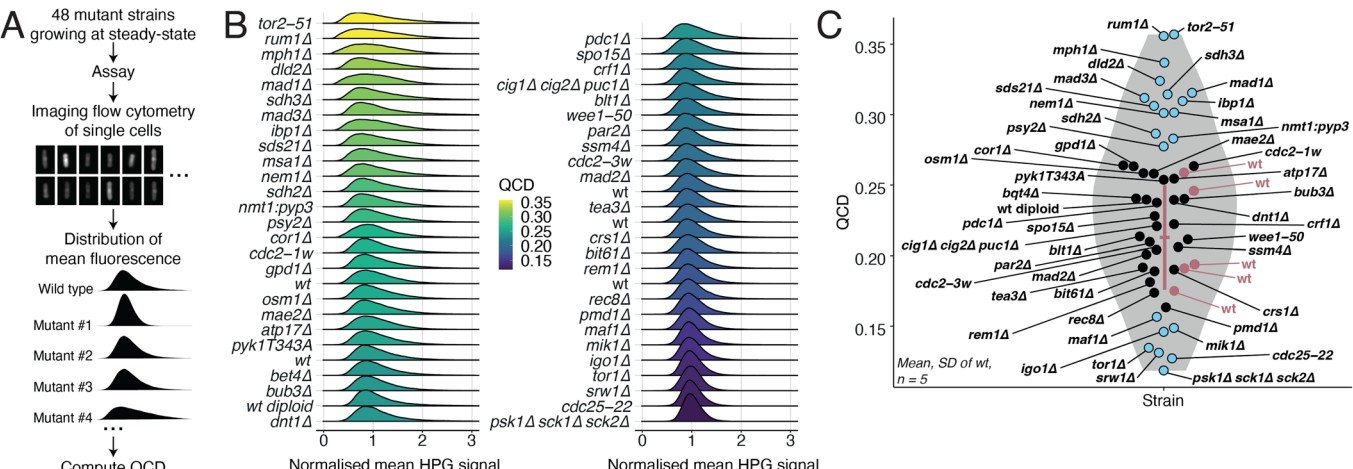

**Figure 3. Genetic modulation of variability in the measured global protein synthesis rate.**

(A) Overview of the variability screen. (B) Kernel density estimates of the normalised mean HPG signal per cell of 48 mutant strains (Table EV1) and 5 wild-type (PN1) experimental replicates. For each strain, the mean HPG signal per cell is divided by the median value for the strain (background not subtracted). (C) QCD of the mean HPG signal per cell for the strains shown in (B). The red bar represents the mean and standard deviation of the five replicates of the wild-type (red dots). Mutants above or below 1.5 standard deviation of the mean wild-type QCD are shown as blue dots, the others are shown as black dots. The shaded area shows the kernel density estimate of all the data. Source data are available online for this figure.

growth conditions and had similar growth rates, with a less than 10% difference compared to wild-type. This resulted in a set of 48 strains with different mutations of genes involved in a range of biological processes (see Table EV1 for the gene list and the associated biological processes). We measured the population QCD of these mutant strains using an imaging flow cytometer (Fig. 3A–C). The red line in Fig. 3C represents the span of variation in the level of variability of five wild-type replicates that was observed with these experiments.

Unexpectedly we found 14 mutants with a population QCD higher than 1.5 standard deviation above the mean QCD of wild-type replicates, and 7 mutants with a QCD lower than 1.5 standard deviation below the mean QCD of wild-type replicates. To confirm the observed high and low variabilities that were found using the imaging flow cytometer, we additionally measured variability using microscopy. We assayed the two mutants with the highest population variability, *rum1Δ* (deletion of a Cyclin-dependent kinase CDK inhibitor) (Moreno and Nurse, 1994) and *tor2-51* (a temperature-sensitive allele of the TORC1 kinase) (Álvarez and Moreno, 2006), and the 2 mutants with the lowest population variability, *tor1Δ* (deletion of the TORC2 kinase) (Otsubo and Yamamato, 2008) and *psk1Δ sck1Δ sck2Δ* (deletion of S6 kinase homologues in the TOR pathway) (Nakashima et al, 2012) (Figs. 4A–F and EV3A–F). The differences in cell-to-cell variability in populations of these mutants were clearly visible in the fluorescence images (Fig. 4A,B). To account for different cell sizes in the population we corrected for the contribution of cell size to the HPG signal (Figs. 4C,D and EV3E). We then calculated the QCD of the length-normalised signal for each population (Fig. 4E,F), and found that the *rum1Δ* and the *tor2-51* populations reproducibly exhibited higher levels of cell-to-cell variability in HPG signal than wild-type populations, whilst the *tor1Δ* and the *psk1Δ sck1Δ sck2Δ* mutants reproducibly exhibited lower levels of variability than wild type. Deleting *sck2* alone but not *psk1* or *sck1*,

resulted in lower variability than wild type (Fig. EV3F). We also found that mutations in regulators of CDK activity had a statistically significant effect on the modulation of variability (Fig. EV4A–F). Overexpression of the protein tyrosine phosphatase Pyp3 (Millar et al, 1992) using the *nmt1* promoter (Rhind and Russell, 2001) increased variability at the population level whilst a partially active mutant of the protein tyrosine phosphatase Cdc25 (Russell and Nurse, 1986) decreased variability compared to wild-type (Fig. EV4A–C). The overexpression of Pyp3, and a mutant of the Cdc2 CDK with increased activity (Nurse and Thuriaux, 1977) led to statistically significantly higher variability than a partially active mutant of the Wee1 protein kinase (Nurse, 1975) (Fig. EV4D–F). Therefore, the variability in HPG signal is genetically influenced, and components of the TOR pathway and of the cell cycle control network are involved in modulation of this variability at the population level. There was almost no detectable correlation ($R^2 = 0.18$) between QCD and the length at division as measured by the mean length of the binucleated cells in the population for the strains we assayed (Fig. EV4G), indicating that differences in mean cell sizes of the different mutants were not responsible for the differences in population heterogeneity.

To understand whether the gene functions of these mutants might reflect multiple parallel pathways affecting variability or if the genes implicated are integrated into one controlling pathway, we generated double deletion strains combining mutations producing high and low cell-to-cell variability in a population and looked for epistatic interactions. Deleting *rum1* in the *tor1Δ* and in the *psk1Δ sck1Δ sck2Δ* backgrounds resulted in populations with low variability (Fig. 4G), suggesting that Tor1 and Sck2 act further downstream than Rum1 in the control pathway of variability. The downstream role of the TOR pathway was further supported by the fact that deleting *tor1* in both *mad1Δ*, and *mad3Δ* backgrounds (deletion of mitotic spindle checkpoint protein genes) (Heinrich et al, 2014), which on their own had exhibited high variability,

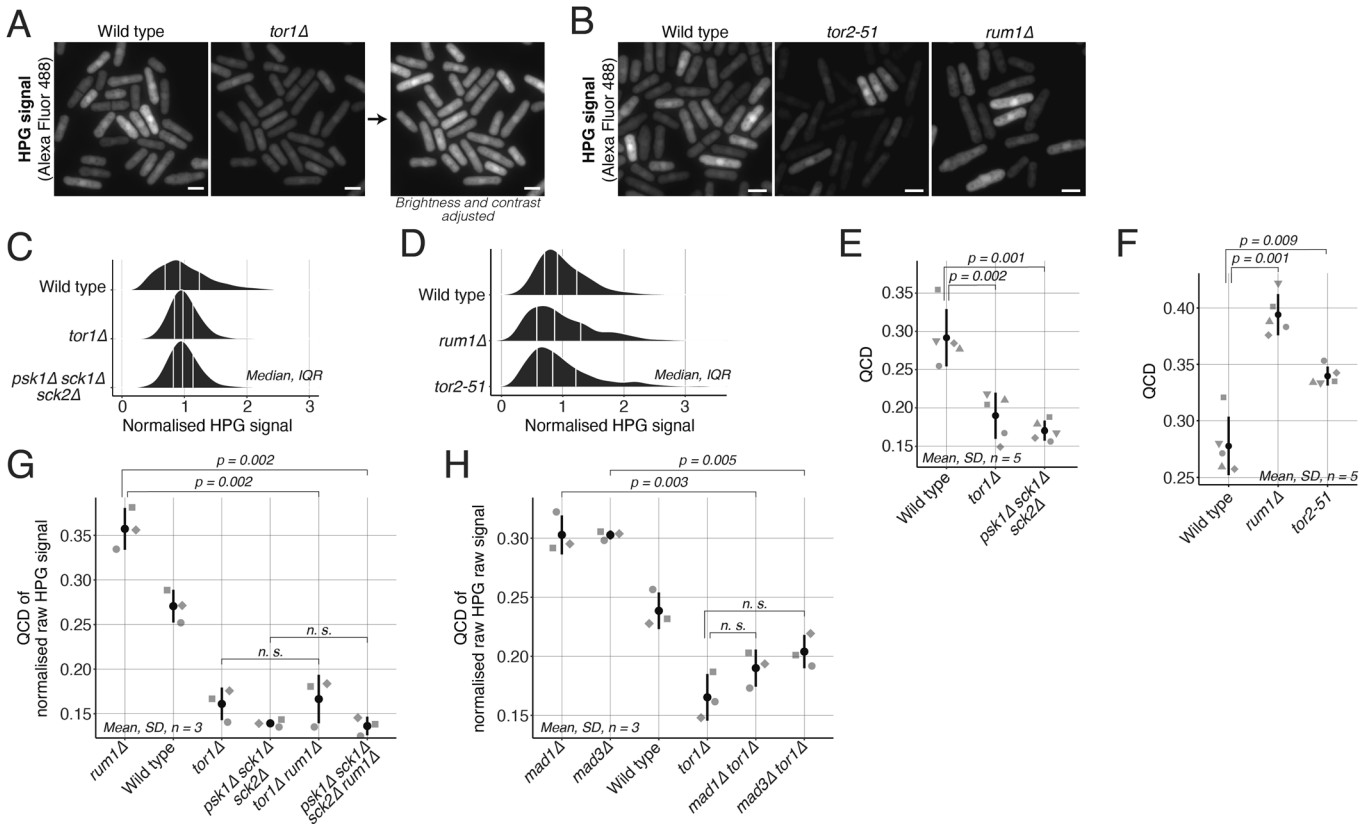

**Figure 4. The TOR pathway and cell cycle controllers modulate variability.**

(A) Example images of HPG signal of wild-type (PN1) and *tor1Δ* (PN5732) cells. The scale bars represent 5 μm. (B) Example images of HPG signal of wild-type (PN1), *tor2-51* (PN5413) and *rum1Δ* (PN957) cells. The scale bars represent 5 μm. (C) Kernel density estimates of the normalised HPG signal of wild-type (PN1), *tor1Δ* (PN5732), and *psk1Δ sck1Δ sck2Δ* (PN5732) populations, calculated using the method shown in Fig. EV3E. The white lines show the first, second, and third quartiles of the population. (D) Same as (C) for wild-type (PN1), *rum1Δ* (PN957), and *tor2-51* (PN5413) populations. (E) Mean and standard deviation of the population QCD of five experimental replicates for the different strains shown in (C). The *P* values are calculated using a two-sided Welch's unequal variances paired *t* test. (F) Same as (E) for the strains shown in (D). (G) Mean and standard deviation of the QCD of the length-normalised raw HPG signal (background not subtracted) for the wild type (PN1), *rum1Δ* (PN957), *tor1Δ* (PN5732), *psk1Δ sck1Δ sck2Δ* (PN5733), *tor1Δ rum1Δ* (PN6080), and *psk1Δ sck1Δ sck2Δ rum1Δ* (PN6081) for three experimental replicates. The single-cell HPG signals and cell length were measured using an imaging flow cytometer. The *P* values are calculated using a two-sided Welch's unequal variances paired *t* test, *P* values above 0.05 are labelled as non-significant (n.s.). (H) Same as (G) for the wild-type (PN1), *mad1Δ* (PN6068), *mad3Δ* (PN6069), *tor1Δ* (PN5732), *mad1Δ tor1Δ* (PN6082), and *mad3Δ tor1Δ* (PN6083). Source data are available online for this figure.

resulted in low-variability populations (Fig. 4H). We conclude that the TOR pathway acts downstream of the cell cycle control network.

## Discussion

We have found that cells of the eukaryote fission yeast growing in steady-state conditions exhibit a significant level of variability in the measured rate of protein synthesis. This was unexpected given that the rate of net protein synthesis is dependent upon the many hundreds of individual reactions and gene functions that are required for amino acid metabolism, translation, and protein degradation. Given the large number of activities involved, it would be expected that they would tend to average each other out at the overall cellular level, but this was not what we observed.

Within an unperturbed, exponentially growing population of fission yeast cells, the measured rate of protein synthesis was highly

variable between cells over a short time scale of 20 min, independent of cell size and cell cycle stage. We observed that cells with rates distant from the mean of the population tend to regress towards the mean population rate within 0.15–0.25 of the generation time, suggesting a homoeostatic mechanism is in operation which corrects deviations from the population mean. We also showed that the variability has an element of genetic determination, with different mutant and gene deletion strains exhibiting increased or decreased variabilities compared with wild type. The fact that the variability in the measured rate of protein synthesis is genetically determined suggests that it is actively regulated, and is not the outcome of an unchangeable intrinsic randomness in the metabolic processes being assayed. This regulatory mechanism results in wild-type cells having a specific level of variability. However, cells can also exhibit both higher and lower levels of variability in particular mutant backgrounds. This indicates that the level of variability found in wild-type cells is set at an optimal level and confers a selective advantage compared with a

more tightly controlled metabolic process that would exhibit less variability, or an even looser control that would exhibit greater variability.

How can these observations be explained? Variability in cellular phenotypic behaviour is usually thought to be due to bet-hedging strategies that can generate a fitness advantage in changing conditions, such as resistance to antibiotic, antifungals or cancer drug treatments (Davidson and Surette, 2008; Bojsen et al, 2017; Inde and Dixon, 2018; Vallette et al, 2019). In populations using bet-hedging strategies, it has been proposed that the optimal long-term growth rate is achieved when the proportion of cells switching between different states mimics the frequency of changes of the fluctuating environment (Müller et al, 2013; Kussell and Leibler, 2005). However, our observation that the measured rate of protein synthesis is highly variable within single cells on short time scales of around 20 min in a steady-state environment is not consistent with this proposal and thus does not support an explanation based on bet-hedging. In addition, it is difficult to imagine how transiently achieving high or low rates of amino acid incorporation into newly synthesised proteins could confer fitness advantage and under what conditions this might be important.

Another more radical alternative to a conventional bet-hedging explanation, is that highly complex metabolic systems such as protein synthesis may need to be more loosely controlled for the entire cellular system to function effectively. There are hundreds of steps between the uptake of amino acids into a cell and their eventual incorporation into newly synthesised proteins. It has been generally assumed that these individual steps and sub-processes are tightly controlled to minimise variability and optimise function. However, this may lead to problems when many such tightly controlled sub-systems have to be integrated together into a complex single system, as is the case for protein synthesis. Therefore there may be advantages if the sub-systems are more loosely connected and not closely controlled together. Tightly regulated and closely connected sub-systems might reduce fluidity and agility in cells slowing down the response of the overall cellular system to both local random perturbations in steady-state growing cells and to changes in environmental circumstances.

Some analogies might be useful for thinking about this. Instability is purposefully built into fighter jets to make them highly responsive and manoeuvrable whilst commercial passenger jets are designed to be resistant to perturbations and thus less manoeuvrable (Stein, 2003; Baetica et al, 2019). In this context it should be noted that the control complexities of a cell far outstrip a jet aircraft. The need for looseness in regulation may be related to the proposal that living cells operate "at the edge of chaos" (Kauffman and Johnsen, 1991) to ensure effective and responsive controls of highly complex and integrated systems operating in a cell. A consequence of this may be the high level of cell-to-cell variability we have observed in the measured rate of protein synthesis. This contrasts with the more usual thinking about controls of metabolism in complex living organisms, which tends to emphasise tightly regulated controls. We propose that there is an optimal level of variability which is a trade-off between allowing sufficient instability to ensure manoeuvrability whilst also maintaining adequate overall stability. Too much stability might lead to an inability to adjust and respond to perturbations, whilst too much instability might lead to these processes being uncontrollable.

Our experiments assaying population heterogeneity in strains with genetic mutations have revealed that variability is altered by mutations affecting cellular processes involving major cellular pathways. It is not evident how many pathways are involved and how they interact to ultimately regulate variability, but the genes we have found to be involved provide clues about how this process might be regulated. The strains exhibiting the most extreme differences in variability of those measured, both high and low, were ones which were deleted or mutated for genes in the TOR pathway and for genes involved in cell cycle control. Epistatic experiments suggested that the TOR pathway acts downstream of cell cycle control.

The master regulator of the cell cycle, CDK (Coudreuse and Nurse, 2010; Swaffer et al, 2016), has previously been implicated in regulation of elements of the TOR pathway through phosphorylation of the translation initiation factor 4E-binding protein (4E-BP1) in human cells (Shuda et al, 2015), and the CDK regulator Cdc25 has been suggested to have a function linking growth rate and the cell cycle (Nurse and Thuriaux, 1984). Since the TOR pathway is a major controller of growth in cells, the Cdc25 function may be acting through the TOR pathway. In addition, a second CDK regulator, Pyp3, has been suggested to physically interact and phosphorylate the translation initiation factor eIF4A Tif1 in fission yeast (Yimit, 2011). These various molecular interactions have the potential to link the modulation of variability in protein synthesis to cell cycle controllers but this will require further investigation.

The TOR pathway is well placed to play a co-ordinating role regulating a range of global cellular metabolic processes ensure fluidity in control and agility in response to change. We have previously shown that around 160 proteins conserved between budding yeast, fission yeast, and mammalian cells undergo rapid changes in their phosphorylation in fission yeast when the TOR pathway is inhibited, which are on time scales comparable with the variability we have observed here in protein synthesis (Mak et al, 2021). These proteins are distributed between major biological processes including signalling, cytoplasmic translation, mRNA metabolic processes, and vesicle-mediated transport. Thus the TOR pathway could act as a link between different parts of global cellular metabolism. Notably, the phosphorylation of many components of the translation machinery, including the translation initiation factors eIF3c Tif303, eIF4E Tif452, eIF4G Tif471, eIF5B Tif52, eIF2ß Tif212, and the eukaryotic translation termination factor 1 Erf1 were found to be subject to TOR regulation indicating that variability in the measured protein synthesis could be regulated through the translation machinery. The TOR pathway has also been involved in many aspects of growth and stress response in yeast and in mammalian cells (Wullschleger et al, 2006; Dibble and Manning, 2013; Álvarez and Moreno, 2006; Laor et al, 2014; van Slegtenhorst et al, 2004; Kawai et al, 2001) including bursts in translation on single mRNA molecules (Livingston et al, 2023). We hypothesise that cells integrate internal cellular and external environmental inputs through the TOR pathway and that elements of the pathway could serve to modulate the level of metabolic variability.

The TOR pathway is conserved throughout eukaryotes so the proposal that TOR may modulate the variability of global cellular processes may be widespread in eukaryotic cells. Data from experiments measuring the rates of protein synthesis at a single-cell level in eukaryotes other than fission yeast including

mammalian cells, are indicative that these rates also exhibit considerable variability (Stonyte et al, 2018; Liu et al, 2021). This possibility combined with the fact that the TOR pathway regulates the phosphorylation of many fission yeast proteins that are also found in other eukaryotes (Mak et al, 2021), is supportive of the idea that a similar fluidity and variability in metabolic processes may be a general organisational feature of living cells, and that the TOR pathway may play a significant role in that mechanism. We propose that this genetic control of fluidity in the global regulation of complex cellular metabolic processes results in a high level of cell variability as we have observed with protein synthesis, and may be a necessary organisational principle of living cells.

## Methods

### Strains

| Species | Genotype | Strain name | Source |
|---|---|---|---|
| S. pombe | 972wt h- | PN1 | Lab collection |
| S. pombe | cdc25-22 h- | PN143 | Lab collection |
| S. pombe | cdc2-1w h- | PN150 | Lab collection |
| S. pombe | wee1-50 h- | PN369 | Lab collection |
| S. pombe | ura4-d18 tor2-51:ura4 h + | PN5413 | Lab collection |
| S. pombe | psk1Δ::hphMX6 sck1Δ::natMX6 sck2Δ::kanMX6 h- | PN5733 | Lab collection |
| S. pombe | ura4-d18 cig1Δ::ura4 cig2Δ::ura4 puc1Δ::ura4 h- | PN5792 | Lab collection |
| S. pombe | leu1-32 mik1Δ::LEU2 h- | PN2341 | Lab collection |
| S. pombe | rec8Δ::kanʳ h- | PN2369 | Lab collection |
| S. pombe | ura-d18 cor1Δ::ura4 h- | PN3007 | Lab collection |
| S. pombe | tea3Δ::kanʳ h- | PN3226 | Lab collection |
| S. pombe | ura4-d18 srw1Δ::ura4 h- | PN3286 | Lab collection |
| S. pombe | ssm4Δ::kan h90 | PN4182 | Lab collection |
| S. pombe | maf1Δ::kanʳ | PN5090 | Lab collection |
| S. pombe | rem1Δ::hphNT h- | PN5342 | Lab collection |
| S. pombe | crs1Δ::hphNT h + | PN5343 | Lab collection |
| S. pombe | pmd1Δ::nat h- | PN5504 | Lab collection |
| S. pombe | blt1Δ::kanMX6 h + | PN10510 | Lab collection |
| S. pombe | dnt1Δ::kanMX6 h- | PN6028 (ER262) | Emma Roberts |
| S. pombe | igo1Δ::natMX6 | PN6052 (JP539) | James Patterson |
| S. pombe | 972wt diploid h-/h- | PN597 | Lab collection |
| S. pombe | rum1Δ:ura | PN957 | Lab collection |
| S. pombe | tor1Δ::kanMX6 | PN5732 | Lab collection |
| S. pombe | msa1Δ::kanMX6 h + | PN6053 (TM58) | Theresa Zeisner |
| S. pombe | pdc1Δ::kanMX6 | PN6054 (TM140) | Theresa Zeisner |
| S. pombe | ibp1Δ::kanMX6 h- | PN6055 (TZ66) | Theresa Zeisner |
| S. pombe | sds21Δ::kanMX6 h- | PN6056 (TZ67) | Theresa Zeisner |
| S. pombe | psy2Δ::kanMX6 h- | PN6057 (TZ80) | Theresa Zeisner |
| S. pombe | par2Δ::kanMX6 h- | PN6058 (TZ88) | Theresa Zeisner |
| S. pombe | mad2Δ::natMX6 cdc2as:bsd | PN6059 (TZ183) | Theresa Zeisner |
| S. pombe | cdc2-3w h- | PN368 | Lab collection |
| S. pombe | ura4-D18 kanMX-P3nmt1:pyp3 cdc25Δ::ura4 h- | PN6060 (CB130) | This paper |
| S. pombe | kanMX-P3nmt1:pyp3 | PN6061 (CB131) | This paper |
| S. pombe | cdc25-22 kanMX-P3nmt1:pyp3 | PN6062 (CB134) | This paper |
| S. pombe | spo15Δ::kanMX6 h- | PN6063 (SW9) | Sarah Willich |
| S. pombe | bqt4Δ::kanʳ h- | PN6064 (SW11) | Sarah Willich |
| S. pombe | crf1Δ::kanMX6 h- | PN6065 (SW19) | Sarah Willich |

| Species | Genotype | Strain name | Source |
|---------|----------|-------------|--------|
| S. pombe | bit61Δ::kanMX6 | PN6066 (SW20) | Sarah Willich |
| S. pombe | mph1Δ::natMX6 | PN6067 (TZ169) | Scott Curran |
| S. pombe | mad1Δ::natMX6 | PN6068 (TZ170) | Scott Curran |
| S. pombe | mad3Δ::natMX6 | PN6069 (TZ171) | Scott Curran |
| S. pombe | bub3Δ::natMX6 | PN6070 (TZ173) | Scott Curran |
| S. pombe | nem1Δ::ura | PN6071 (SO8122) | Oliferenko Lab collection |
| S. pombe | dld2Δ::kan$^r$ h- | PN6072 (SO8563) | Oliferenko Lab collection |
| S. pombe | sdh2Δ::kan$^r$ h- | PN6073 (SO8685) | Oliferenko Lab collection |
| S. pombe | atp17Δ::kan$^r$ h- | PN6074 (SO8691) | Oliferenko Lab collection |
| S. pombe | sdh3Δ::kan$^r$ h- | PN6075 (SO8695) | Oliferenko Lab collection |
| S. pombe | mae2Δ::kan$^r$ h- | PN6076 (SO8724) | Oliferenko Lab collection |
| S. pombe | osm1Δ::kan$^r$ h- | PN6077 (SO8728) | Oliferenko Lab collection |
| S. pombe | gpd1Δ::hph | PN6078 (SO8991) | Oliferenko Lab collection |
| S. pombe | pyk1T343A | PN6079 | Oliferenko Lab collection, from Kamrad et al (Kamrad et al, 2020) |
| S. pombe | tor1Δ::kanMX6 rum1Δ::blaMX6 | PN6080 (CB182) | This paper |
| S. pombe | psk1Δ::hphMX6 sck1Δ::natMX6 sck2Δ::kanMX6 rum1Δ::blaMX6 | PN6081 (CB184) | This paper |
| S. pombe | tor1Δ::hphMX6 mad1Δ::natMX6 | PN6082 (CB206) | This paper |
| S. pombe | tor1Δ::hphMX6 mad3Δ::natMX6 | PN6083 (CB207) | This paper |
| S. pombe | 975wt h + | PN4 | Lab collection |
| S. pombe | ura4-D18 h + | PN731 | Lab collection |
| S. pombe | leu1-32 ura4-D18 kanMX-P3nmt1:pyp3 cdc25Δ::ura4 h + | PN6051 (NR2613) | Paul Russell (Rhind and Russell, 2001) |
| S. pombe | psk1Δ::hphMX6 h- | PN6084 (TM55, CB152) | Tiffany Mak |
| S. pombe | sck1Δ::natMX6 h- | PN6085 (TM53, CB153) | Tiffany Mak |
| S. pombe | sck2Δ::kanMX6 h- | PN6086 (TM57, CB154) | Tiffany Mak |

## Oligonucleotides

| Name | Sequence | Source |
|------|----------|--------|
| d124_rum1del_F | TACACGCTCTTCTGATAATCCGCTCGATACATAAAGGATTTCTTCTTGCAT TTACCTGGTTTTTGGATTGTCAGTTCGCTCGGATCCCCGGGTTAATTAA | Joseph Curran |
| d125_rum1del_R | ATGAATAAGGCAGAAGAGTATTTCGTGATTGGGCATTTATATAAACGGTAT CAAACACAATTACAAAATGCGAAAAAAAGGAATTCGAGCTCGTTTAAAC | Joseph Curran |
| dpTM66 | TCCGGTCAACCTAATACTGAATTGTTTGCTTTCCGTAGACATTGTGATGAA TGCCTAAGTGGAAGAATTGAACACCGCGACGGATCCCCGGGTTAATTAA | Tiffany Mak |
| dpTM67 | GAGAAGTCTCTTTGAAATTTTTGATGAGTATGAGAAATAAAATAGTCAT CCAGGAAAAGAATCATAACTTATTTGGAGCTGAATTCGAGCTCGTTTAAAC | Tiffany Mak |

## Strain construction

Strains were constructed using previously described procedures (Moreno et al, 1991). PN6061 was constructed using random spore analysis after a genetic cross between PN6051 and PN4. PN6060 was constructed using random spore analysis after a genetic cross between PN6051 and PN731. PN6062 was constructed using random spore analysis after a genetic cross between PN143 and PN6061. PN6080 and PN6081 were constructed using the lithium acetate transformation method (Moreno et al, 1991) with PN5732 and PN5733, respectively, with the PCR product of a pFA6a series plasmid carrying a *bsdMX6* cassette (lab collection), and the d124_rum1del_F and d125_rum1-del_R primers. PN6082 and PN6083 were constructed using the lithium acetate transformation method with PN6068 and PN6069, respectively, with the PCR product of a pFA6a series plasmid carrying a *hphMX6* cassette (lab collection), and the dpTM66 and the dpTM67 primers. All genotypes were confirmed phenotypically when possible (for conditionally lethal markers) and by PCR for gene deletions.

## Cell culture

Cells were cultured following previously described procedures (Moreno et al, 1991). Frozen stocks ($-80\,°C$) of stationary cultures of cells in Yellow Freezing Mix (YFM: 5 g L$^{-1}$ Yeast Extract; 30 g L$^{-1}$ Glucose; 250 mg L$^{-1}$ Histidine; 250 mg L$^{-1}$ Leucine; 250 mg L$^{-1}$ Adenine; 250 mg L$^{-1}$ Lysine; 250 mg L$^{-1}$ Uridine; 250 mg L$^{-1}$ Glutamic Acid, 250 ml L$^{-1}$ Glycerol) were streaked on Yeast Extract Agar plates (YE4S: 5 g L$^{-1}$ Yeast Extract; 30 g L$^{-1}$ Glucose; 150 mg L$^{-1}$ Histidine; 150 mg L$^{-1}$ Leucine; 150 mg L$^{-1}$ Adenine; 150 mg L$^{-1}$ Uridine; 20 g L$^{-1}$ Agar) and incubated at $25\,°C$ for 72 h. Single colonies were patched on Edinburgh minimal medium Agar plates (EMM, MP Biomedicals: 3 g L$^{-1}$ Potassium Hydrogen Phthalate; 2.2 g L$^{-1}$ Na$_2$HPO$_4$; 5 g L$^{-1}$ NH$_4$Cl; 20 g L$^{-1}$ Glucose; 1.05 g L$^{-1}$ MgCl$_2$·6H$_2$O; 14.7 µg L$^{-1}$ CaCl$_2$·2H$_2$O; 1 g L$^{-1}$ KCl; 0.04 g L$^{-1}$ Na$_2$SO$_4$; 1 mg L$^{-1}$ Pantothenic acid; 10 mg L$^{-1}$ Nicotinic Acid; 10 mg·L$^{-1}$ Myo-inositol; 10 µg L$^{-1}$ Biotin; 0.5 mg L$^{-1}$ Boric Acid; 0.4 mg L$^{-1}$ MnSO$_4$; 0.4 mg L$^{-1}$ ZnSO$_4$·7H$_2$O; 0.2 mg L$^{-1}$ FeCl$_2$·6H$_2$O; 40 µg L$^{-1}$ Molybdic Acid; 0.1 mg L$^{-1}$ KI; 40 µg L$^{-1}$ CuSO$_4$·5H$_2$O; 1 mg L$^{-1}$ Citric Acid; 20 g L$^{-1}$ Agar) and incubated at $25\,°C$ for 24 h. Single patches were inoculated into 5 ml of EMM and incubated at $25\,°C$ overnight in a stationary incubator. The cultures were then diluted in EMM to OD$_{595}$ = 0.05 (calculated using an Amersham Ultraspec 2100 pro) in a flask and incubated at $25\,°C$ in a shaking incubator for 8 h. The cultures were then diluted in EMM to OD$_{595}$ = 0.25 in a flask and incubated at $25\,°C$ in a shaking incubator overnight before being diluted for the experiment.

## Exogenous methionine analogue incorporation assay

Cells were assayed following the protocol we previously described (Basier and Nurse, 2023). An exponentially growing culture was diluted to OD$_{595}$ = 0.3 in 20 ml EMM at $25\,°C$ in a 50-ml flask, and placed in a shaking water bath for 1 h. Next, 4 µl of HPG (Cambridge Bioscience) was added to the culture from a 50 mM stock solution in Milli-Q water to a final concentration of 10 µM. Immediately after the addition of HPG, a 3.84 ml sample of the culture was taken and fixed with 1.16 ml of a stock solution of 16% (w/v) formaldehyde (methanol-free, Thermo Fisher) in a 15-ml centrifuge tube, to a final concentration of 3.7%, and vortexed for 5 s before being incubated at room temperature on a rocker, in the dark, for 40 min. This first sample was used to compute the background signal. After 5 min, a second sample was taken from the culture and processed the same way, apart from being incubated for only 30 min. Fixed cells were then spun at 2000 rcf for 5 min, and the supernatant was discarded. Cells were resuspended in 3 ml of PBS (Gibco) + 1% (w/v) BSA (Sigma-Aldrich), vortexed for 5 s, spun at 2000 rcf for 5 min, and the supernatant was discarded. Cells were resuspended in 6 ml of PBS + 1% (w/v) BSA + 1% (v/v) Triton X-100 (Sigma-Aldrich), vortexed for 5 s, and incubated at room temperature on a rocker for 30 min, in the dark. Cells were spun at 2000 rcf for 5 min, the supernatant was discarded, and cells were resuspended in 6 ml of PBS + 1% (w/v) BSA, vortexed for 5 s and incubated at room temperature on a rocker for 60 min, in the dark. Cells were spun at 2000 rcf for 5 min, the supernatant was discarded, and cells were resuspended in 500 µl of 1× Click-iT reaction buffer (Thermo Click-iT Plus picolyl azide kit, Thermo Fisher), and transferred to a 1.5-ml centrifuge tube. Cells were spun at 2000 rcf for 5 min, the supernatant was discarded, and resuspended in 500 µl of the following reaction mix from the Thermo Click-iT Plus picolyl azide kit: 870 µl of 1X Click-iT reaction buffer (A), 10 µl of Alexa fluor at 500 µM (B), 15 µl of CuSO$_4$ at 100 mM (C), 5 µl of Copper protectant (D), 10 µl of 10× Click-iT buffer additive (E), 90 µl of Milli-Q water (F). To make the reaction mix, the solutions

were added in the following order: A was mixed with B, C was mixed with D, E was mixed with F, AB was mixed with CD, EF was mixed with ABCD. Cells were incubated at room temperature on a shaker at 1000 rpm for 30 min in the dark. Cells were then spun at 17,000 rcf for 15 s, the supernatant was discarded, and cells were resuspended in 800 µl of 50 mM sodium citrate (Fisher Bioreagents) and vortexed for 5 s. Cells were spun at 17,000 rcf for 15 s, the supernatant was discarded, cells were resuspended in 800 µl of 50 mM sodium citrate + 1:10,000 Nuclear-ID Blue (Enzo Life Sciences), and vortexed for 5 s. Cells were spun at 17,000 rcf for 15 s, the supernatant was discarded, cells were resuspended in 800 µl of 50 µl sodium citrate. Cells were spun at 17,000 rcf for 1 s, the supernatant was discarded, cells were resuspended in 500 µl of 50 mM sodium citrate, and stored at $4\,°C$ in the dark for 1 h before imaging.

## Dual pulse experiments

Wild-type cells (PN1) were incubated with 10 µM HPG for 10 min, washed three times and resuspended in medium without HPG, 20 µM AHA (Thermo Fisher) from a 50 mM stock solution in Milli-Q water was added 10 min before fixation. HPG and AHA were either added simultaneously or AHA was added 20, 50, or 110 min after HPG. Cells were stained with 50 µM Alexa Fluor azide 488, washed in 1× Click-iT reaction buffer, then stained with 50 µM Alexa Fluor 647 alkyne.

## Microscopy setup, image segmentation, and quantification

All bright-field and fluorescence microscopy were performed using a Nikon Eclipse Ti2 inverted microscope equipped with a Nikon Perfect Focus System, Okolab environmental chamber and a Photometrics Prime Scientific CMOS camera. The microscope was controlled using the Micro-Manager v2.0 software. Fluorescence excitation was performed with a Lumencor Spectra X light engine fitted with the following excitation filters; 395/25 nm for imaging Nuclear-ID Blue; 470/24 nm for imaging Alexa Fluor 488; and 640/30 nm for imaging Alexa Fluor 647. The emission filters used were the following: Semrock Brightline 438/24 nm for imaging Nuclear-ID Blue, Chroma ET525/50 m for imaging Alexa Fluor 488; and Semrock Brightline 680/42 nm for imaging Alexa Fluor 647. The dichroic mirrors used were the following: Semrock 409/493/573/652 nm BrightLine quad-edge standard epi-fluorescence dichroic beam splitter for imaging Nuclear-ID Blue, Alexa Fluor 488 and Alexa Fluor 647. Images were taken using a Nikon Plan Apo 100×/1.45 Lambda oil immersion objective.

The bright-field images 1 µm below the focal plane of cells have a distinct outline and were therefore used to generate whole-cell masks using Ilastik-1.3.0-OSX. The cell masks generated this way overlap well with the cells on the focal plane images.

The fluorescence images of the focal plane and the $\pm 0.5$ µm z-stacks were maximum projected, and all subsequent analysis was done on the maximum projected fluorescence images. To generate the DNA masks, Ilastik-1.3.0-OSX (Berg et al, 2019) was used on the Nuclear-ID Blue fluorescence images. To determine the number of nuclei per cell, the number of DNA masks within each whole-cell mask was calculated using Fiji (ImageJ version 2.1.0/1.53c). On all images, the scale was set using the function Analyse > Set Scale of Fiji so that the distance between 15.3609 pixels corresponded to 1 µm. To generate single-cell measurements of cell length, the Analyse > Analyse particles function of Fiji was used on the whole-

cell masks to calculate for each mask; its Feret's diameter (the longest distance between any two points within a mask used as a measurement of cell length), and its area. Then, the Analyse > Analyse particles function was used with the cell masks to calculate their corresponding fluorescence measurements on the fluorescence images, comprising the total pixel intensity, mean pixel intensity, median pixel intensity and maximum pixel intensity. The masks were indexed so that the single-cell measurements of the different channels and the measurement of the number of nuclei were attributed to their corresponding cell mask. The data were then processed using R (version 4.1.0) and RStudio (version 1.4.1106). The median total fluorescence intensity of the background sample within an experiment (cells immediately fixed after the addition of HPG) was calculated. Then, the total fluorescence intensity of each cell was divided by the median background total fluorescence intensity. This allows all experiments to have fluorescence values roughly on the same scale and is convenient for processing. Next, the background was subtracted based on cell length. Cells are grouped based on their length in bins spanning 1 μm. For each length bin, the median background total fluorescence intensity was calculated on the background samples and subtracted from each cell's total fluorescence intensity according to its length. The resulting value was used as a measure of HPG incorporation. Binucleated cells were excluded from the analysis.

### Imaging flow cytometry

Cells were assayed for exogenous methionine analogue incorporation and resuspended in 60 μl of 50 mM sodium citrate, water bath sonicated for 30 s using a JSP Digital Ultrasonic Cleaner, then imaged for brightfield and fluorescence (488 nm laser, bandpass 480–560 nm for imaging Alexa Fluor 488 and laser 405 nm, bandpass 420–505 nm for imaging Nuclear-ID Blue) with an Amnis Imagestream X Mk II Imaging Flow Cytometer with a 60× objective lens. Prior to acquisition, cells were gated based upon Brightfield Gradient RMS values (value of cell focus) of 62 to 85 and Area vs Aspect Ratio values consistent with single cells. Approximately 100,000 gated cells were recorded. Post-acquisition processing was made with Amnis IDEAS 6.2 software. Cell segmentation masks were created from bright-field images: Erode(MO1, 3). Single cells were further isolated from doublets and debris using the Thickness Min, Thickness Max, and Width of the segmentation mask. Mononucleated cells were isolated using the Major Axis Intensity of the 420–505 nm channel. The mean HPG signal per cell was calculated using the segmentation mask on the 480–560 nm channel.

### Length-normalisation of HPG signal

An OLS linear regression was fitted on the HPG signal as a function of cell length in the population. Then, for each cell, the experimentally observed HPG signal was divided by the expected signal corresponding to its cell length predicted by the OLS linear regression.

### Graphs and statistics

All plots and statistical analyses were performed using R (version 4.1.0) and RStudio (version 1.4.1106). Statistical analyses used and *n* numbers are stated in the figures and figure captions.

## Data availability

This study includes no data deposited in external repositories.

## Peer review information

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

## Acknowledgements

The authors thank J Curran, J Greenwood, B Whyte, T Hammond, and N Kapadia for their comments on the manuscript, T Mak for the early discussions, and Ryoko Mandeville for laboratory support. We thank the laboratory of S Oliferenko as well as P Russell, S Willich, T Zeisner, T Mak, S Curran, J Patterson, and E Roberts for sharing their strains. We also thank the Flow Cytometry and the Advanced Light Microscopy facilities at the Francis Crick Institute for their help with imaging flow cytometry and microscopy, respectively. This work was supported by the Francis Crick Institute which receives its core funding from Cancer Research UK (CC2003), the UK Medical Research Council (CC2003), and the Wellcome Trust (CC2003). In addition, this work was supported by the Wellcome Trust Grant to PN [grant number 214183 and 093917], The Lord Leonard and Lady Estelle Wolfson Foundation, Woosnam Foundation and Breast Cancer Research Foundation [BCRF-22-117]. For the purpose of Open Access, the author has applied a CC BY public copyright licence to any Author Accepted Manuscript version arising from this submission.

## Author contributions

**Clovis Basier**: Conceptualisation; Data curation; Formal analysis; Validation; Investigation; Visualisation; Methodology; Writing—original draft; Project administration; Writing—review and editing. **Paul Nurse**: Conceptualisation; Formal analysis; Supervision; Funding acquisition; Writing—original draft; Writing—review and editing.

## Funding

## Disclosure and competing interests statement

The authors declare no competing interests. Paul Nurse is a member of the Advisory Editorial Board of *The EMBO Journal*. This has no bearing on the editorial consideration of this article for publication.

# Expanded View Figures

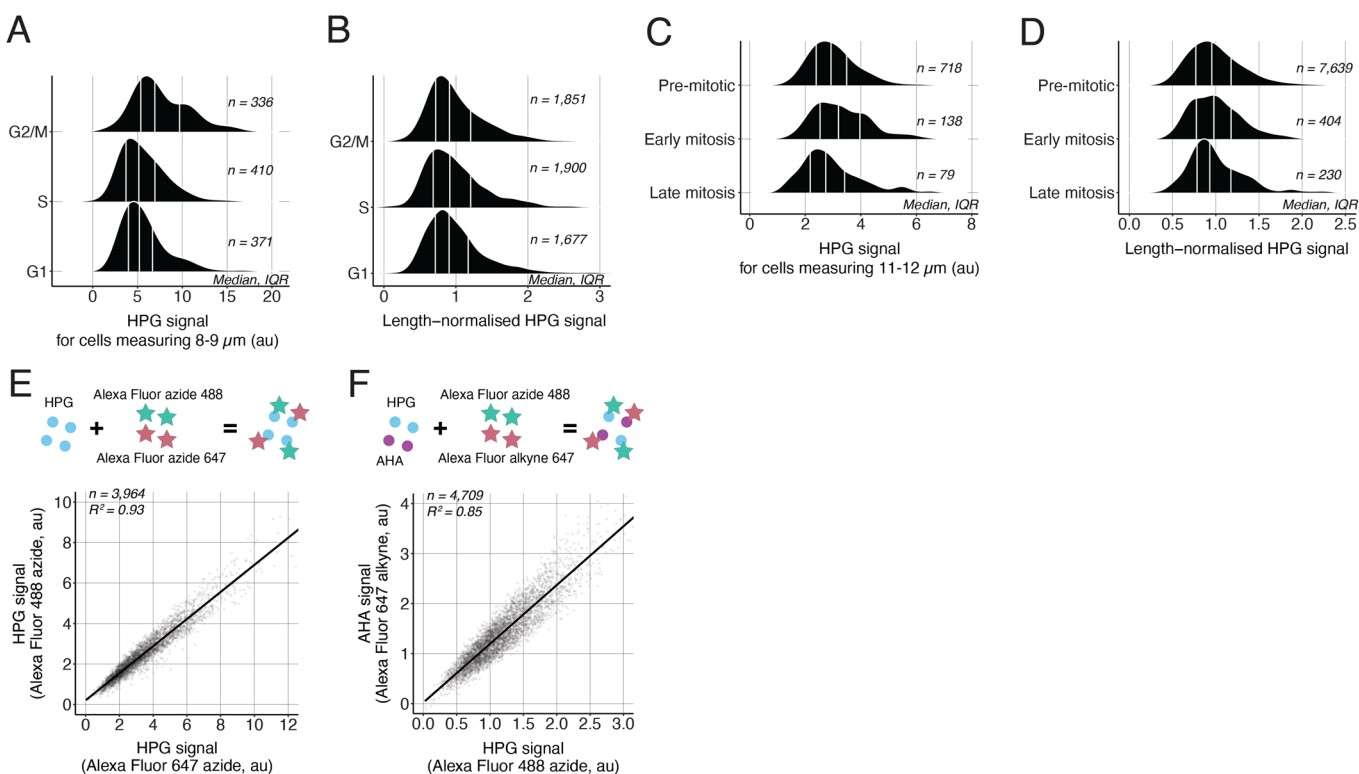

**Figure EV1.  Influence of the cell cycle stage and the staining procedure on the distribution of the HPG signal.**

(**A**) Previously published data of an asynchronous population of *cig1Δ cig2Δ puc1Δ EGFP-pcn1* cells (PN6001) assayed with HPG (Basier and Nurse, 2023) was used to assess the influence of the G1, S, and G2/M cell cycle stages on the distribution of HPG signals. Kernel density estimates of the HPG signal of cells with lengths between 8 and 9 µm. The white lines show the first, second, and third quartiles of the population. (**B**) Kernel density estimates of the length-normalised HPG signal (using the transformation described in Fig. EV3E) for all the *cig1Δ cig2Δ puc1Δ EGFP-pcn1* cells (PN6001) identified as G1, S, or G2/M in the previously published data (Basier and Nurse, 2023). The white lines show the first, second, and third quartiles of the population. (**C**) Previously published data of an asynchronous population of *synCut3-mCherry* cells (PN6004) assayed with HPG (Basier and Nurse, 2023) was used to assess the influence of mitosis on the distribution of HPG signals. Kernel density estimates of the HPG signal of cells with lengths between 11 and 12 µm. The white lines show the first, second, and third quartiles of the population. Pre-mitotic, early mitosis, and late mitosis classifications correspond to "uninucleate, low nuclear synCut3", "uninucleate, high nuclear synCut3", and "binucleate, high nuclear synCut3" in the previously published data, respectively. (**D**) Same as (**B**) for all the *synCut3-mCherry* cells (PN6004) classified in the previously published data (Basier and Nurse, 2023). (**E**) Single-cell measurements of cells incubated for 5 min with 10 µm HPG and co-stained with 2.5 µM Alexa Fluor azide 488 and 2.5 µM Alexa Fluor azide 647 (black dots). The black line is the OLS linear regression fitted on the single-cell data. (**F**) Single-cell measurements of cells co-incubated with 10 µM HPG and 20 µM AHA, and stained with Alexa Fluor azide 488 and Alexa Fluor alkyne 647 (black dots). The black line is the OLS linear regression fitted on the single-cell data.

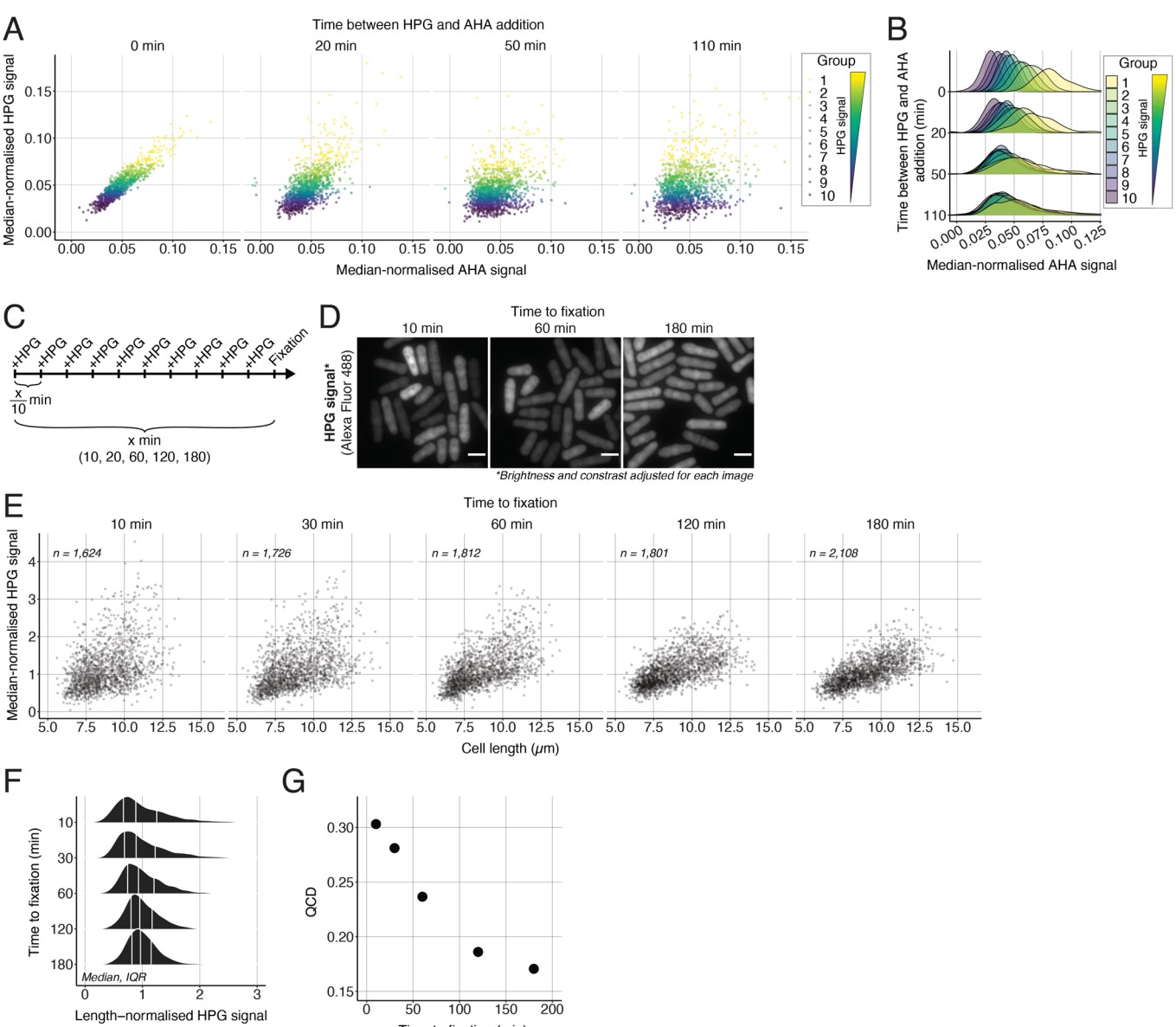

**Figure EV2. Dynamics of single-cell variability in the measured global protein synthesis rate.**

(A) The single-cell data shown in Fig. 2B is colour-coded according to the group attributed to each cell in Fig. 2E–G. (B) Same as Fig. 2F for all ten groups. (C) Schematic of the repetitive HPG pulses experiment. Wild-type (PN1) cells were incubated with HPG for 10, 30, 60, 120, or 180 min. Each incubation was divided in ten intervals of the same length and 1 µM HPG was added at the beginning of each interval. (D) Example images of HPG signal for the 10-, 60-, and 180-minute incubations. The scale bars represent 5 µm. (E) For each incubation shown in (A), the single-cell HPG signal is divided by the median signal of the incubation. (F) Kernel density estimate of the length-normalised HPG signal (using the transformation described in Fig. EV3E) of the populations shown in (E). The white lines show the first, second, and third quartiles of the population. (G) QCDs of the length-normalised HPG signal distributions shown in (F).

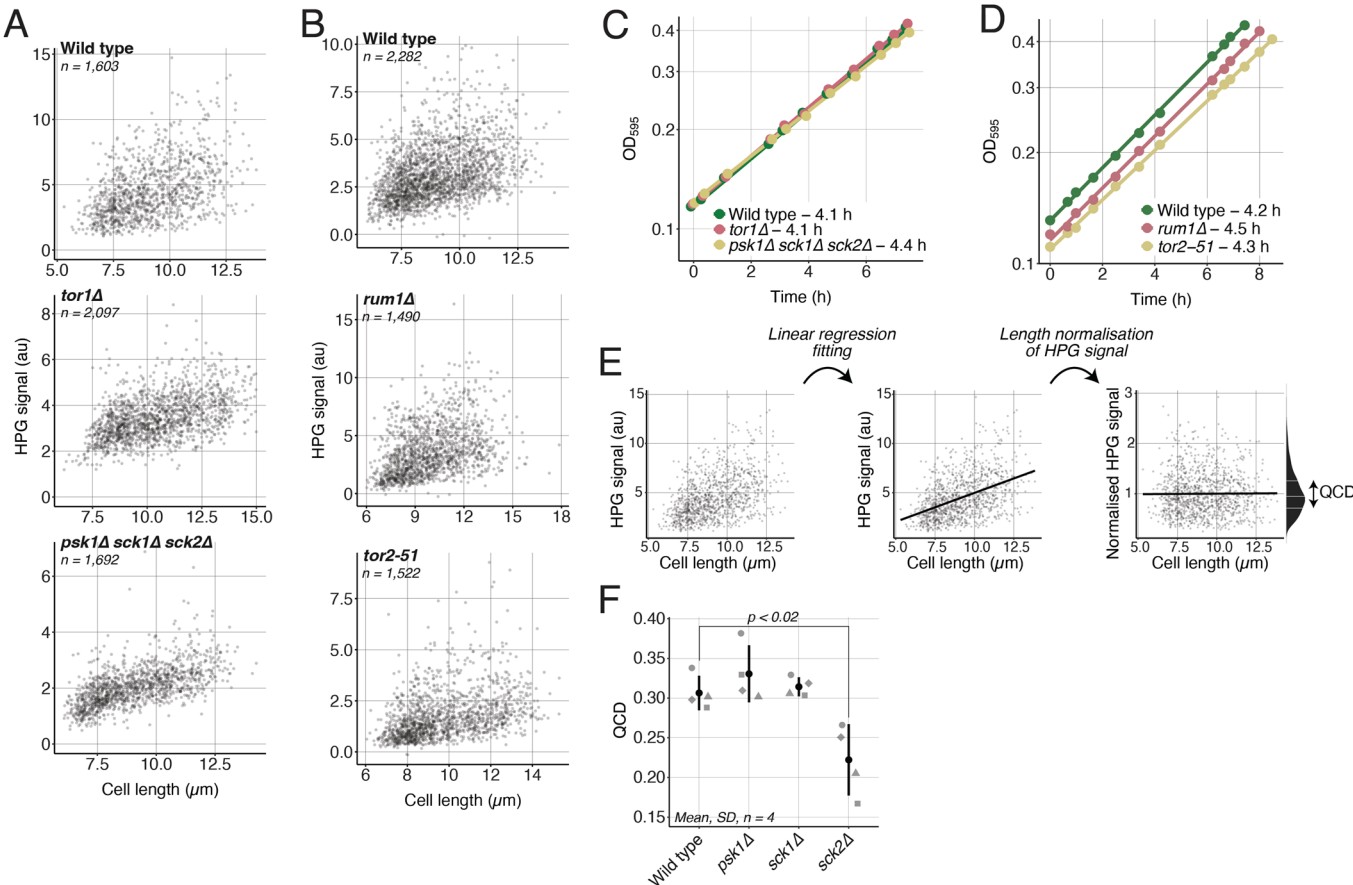

**Figure EV3. The TOR pathway and cell cycle controllers modulate variability.**

(A) Single-cell HPG signal in exponentially growing wild-type fission yeast (PN1), *tor1Δ* (PN5732), and *psk1Δ sck1Δ sck2Δ* (PN5733) populations of cells. (B) Same as (A) for wild type (PN1), *rum1Δ* (PN957), and *tor2-51* (PN5413) populations. (C) Growth curves of the different strains shown in (A). The doubling time of each strain indicated in the legend is calculated using the slope the OLS linear regression fitted on the data after a natural logarithmic transformation. $OD_{595}$ stands for optical density at 595 nm. (D) Same as (C) for the strains shown in (B). (E) Schematic of the data processing method used to compute the population QCD after removing the contribution of cell length to the measured variability. An OLS linear regression is fitted on the HPG signal as a function of cell length (black line on the middle panel), then for each cell the normalised HPG signal is obtained by dividing the observed HPG signal by the expected HPG signal for that cell length computed using the OLS linear regression. The distribution to the right of the right panel is the kernel density estimate of the single-cell normalised HPG signal used to compute the QCD. The white lines show the first, second, and third quartiles of the population. (F) Mean and standard deviation of the population QCD of four experimental replicates wild type (PN1), *psk1Δ* (PN6084), *sck1Δ* (PN6085), and *sck2Δ* (PN6086). The p values are calculated using a two-sided Welch's unequal variances paired *t* test.

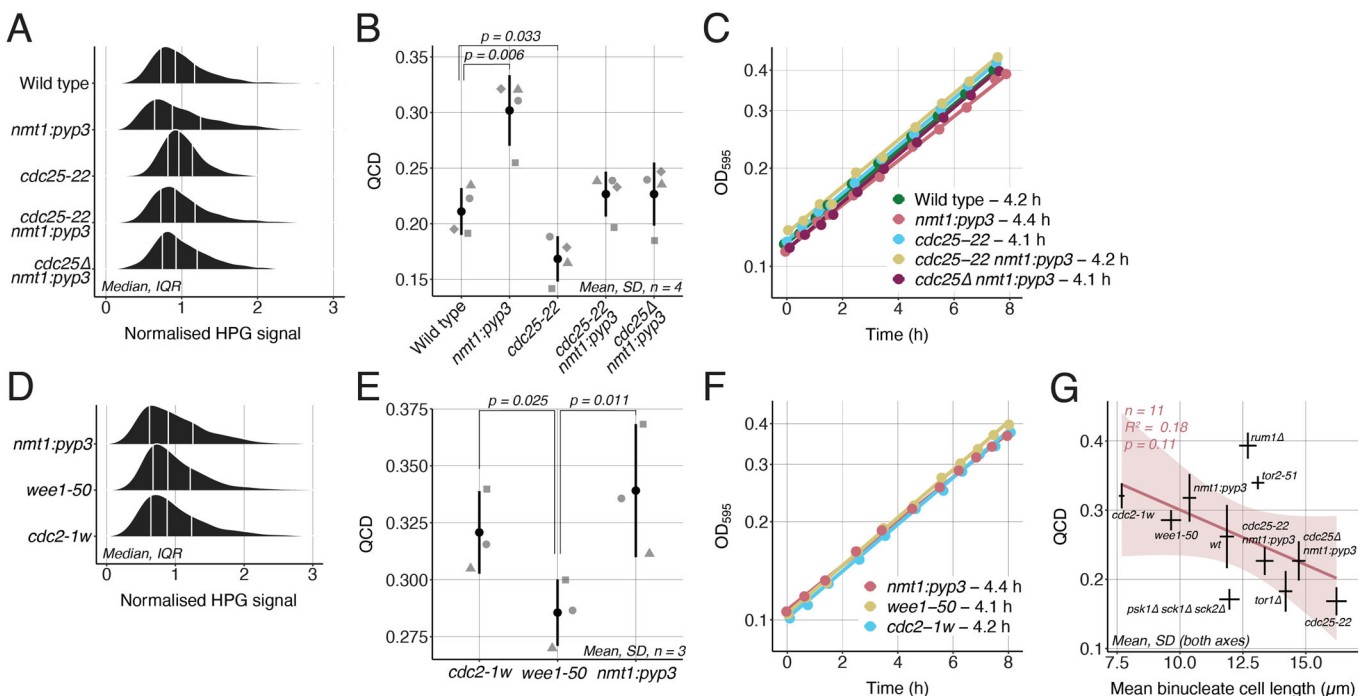

**Figure EV4. Cell cycle controllers influence variability.**

(A) Kernel density estimates of the normalised HPG signal of wild type (PN1), *nmt1:pyp3* (PN6061), *cdc25-22* (PN143), *cdc25-22 nmt1:pyp3* (PN6062), and *cdc25Δ nmt1:pyp3* (PN6060) populations, calculated using the method shown in Figure EV3E. The white lines show the first, second, and third quartiles of the population. (B) Mean and standard deviation of the population QCD of four experimental replicates for the different strains shown in (A). The *P* values are calculated using a two-sided Welch's unequal variances paired *t* test. (C) Growth curves of the different strains shown in (A, B). The doubling time of each strain indicated in the legend is calculated using the slope the OLS linear regression fitted on the data after a natural logarithmic transformation. $OD_{595}$ stands for optical density at 595 nm. (D) Same as (A) for *cdc2-1w* (PN150), *wee1-50* (PN369), and *nmt1:pyp3* (PN6061) populations. (E) Same as (B) for the strains shown in (D). (F) Same as (C) for the strains shown in (D, E). (G) QCD as a function of the mean binucleate cell length in the population for the experimental replicates shown in (B), (F), and Fig. 4E, F. Mean and standard deviation shown for both axes. For strains appearing on multiple panels, all the experimental replicates are pooled together. The red line represents the OLS linear regression fitted on the mean values for each strain, the shaded area indicates the 95% confidence interval.

