## [Peer Review File · The EMBO Journal]

TOR regulates variability of protein synthesis rates

Clovis Basier and Paul Nurse

Corresponding author: Clovis Basier (clovis.basier@crick.ac.uk)

Review Timeline:

Submission Date:	23rd Oct 23
Editorial Decision:	22nd Nov 23
Revision Received:	23rd Jan 24
Editorial Decision:	6th Feb 24
Revision Received:	20th Feb 24
Accepted:	23rd Feb 24

Editor: Daniel Klimmeck

Transaction Report:

Dear Dr Basier,

Thank you again for submitting your manuscript for consideration by the EMBO Journal. Please accept my apologies for getting back to you with protraction due to delayed referee input, as well as detailed discussion in the editorial team. Your manuscript has been seen by three referees, and we have received reports from all of them, which are shown below.

Given the referees' positive recommendations, I would like to invite you to submit a revised version of the manuscript, addressing the comments of all three reviewers. I should add that it is EMBO Journal policy to allow only a single round of revision, and acceptance of your manuscript will therefore depend on the completeness of your responses in this revised version.

I would appreciate if you could contact me during the next weeks for exchange e.g. a video call to discuss your perspective on the comments and potential plan for revisions.

Please feel free to contact me if you have any questions or need further input on the referee comments.

When submitting your revised manuscript, please carefully review the instructions below.

Please feel free to approach me any time should you have additional questions related to this.

Thank you for the opportunity to consider your work for publication.

I look forward to your revision.

Kind regards,

Daniel Klimmeck

Daniel Klimmeck, PhD
Senior Editor
The EMBO Journal

Instruction for the preparation of your revised manuscript:

- 1) a .docx formatted version of the manuscript text (including legends for main figures, EV figures and tables). Please make sure that the changes are highlighted to be clearly visible.
- 2) individual production quality figure files as .eps, .tif, .jpg (one file per figure).
- 3) a .docx formatted letter INCLUDING the reviewers' reports and your detailed point-by-point response to their comments. As part of the EMBO Press transparent editorial process, the point-by-point response is part of the Review Process File (RPF), which will be published alongside your paper.
- 4) a complete author checklist, which you can download from our author guidelines ([https://wol-prod-cdn.literatumonline.com/pb-assets/embo-site/Author Checklist%20-%20EMBO%20J-1561436015657.xlsx](https://wol-prod-cdn.literatumonline.com/pb-assets/embo-site/Author%20Checklist%20-%20EMBO%20J-1561436015657.xlsx)). Please insert information in the checklist that is also reflected in the manuscript. The completed author checklist will also be part of the RPF.
- 5) Please note that all corresponding authors are required to supply an ORCID ID for their name upon submission of a revised manuscript.
- 6) It is mandatory to include a 'Data Availability' section after the Materials and Methods. Before submitting your revision, primary

datasets produced in this study need to be deposited in an appropriate public database, and the accession numbers and database listed under 'Data Availability'. Please remember to provide a reviewer password if the datasets are not yet public (see <https://www.embopress.org/page/journal/14602075/authorguide#datadeposition>).

7) Our journal encourages inclusion of *data citations in the reference list* to directly cite datasets that were re-used and obtained from public databases. Data citations in the article text are distinct from normal bibliographical citations and should directly link to the database records from which the data can be accessed. In the main text, data citations are formatted as follows: "Data ref: Smith et al, 2001" or "Data ref: NCBI Sequence Read Archive PRJNA342805, 2017". In the Reference list, data citations must be labelled with "[DATASET]". A data reference must provide the database name, accession number/identifiers and a resolvable link to the landing page from which the data can be accessed at the end of the reference. Further instructions are available at .

8) At EMBO Press we ask authors to provide source data for the main and EV figures. Our source data coordinator will contact you to discuss which figure panels we would need source data for and will also provide you with helpful tips on how to upload and organize the files.

Numerical data can be provided as individual .xls or .csv files (including a tab describing the data). For 'blots' or microscopy, uncropped images should be submitted (using a zip archive or a single pdf per main figure if multiple images need to be supplied for one panel). Additional information on source data and instruction on how to label the files are available at .

9) We replaced Supplementary Information with Expanded View (EV) Figures and Tables that are collapsible/expandable online (see examples in <https://www.embopress.org/doi/10.15252/emj.201695874>). A maximum of 5 EV Figures can be typeset. EV Figures should be cited as 'Figure EV1, Figure EV2' etc. in the text and their respective legends should be included in the main text after the legends of regular figures.

11) For data quantification: please specify the name of the statistical test used to generate error bars and P values, the number (n) of independent experiments (specify technical or biological replicates) underlying each data point and the test used to calculate p-values in each figure legend. The figure legends should contain a basic description of n, P and the test applied. Graphs must include a description of the bars and the error bars (s.d., s.e.m.).

We realize that it is difficult to revise to a specific deadline. In the interest of protecting the conceptual advance provided by the work, we recommend a revision within 3 months (20th Feb 2024). Please discuss the revision progress ahead of this time with the editor if you require more time to complete the revisions.

Referee #1:

The manuscript by Basier and Nurse identifies high variability in protein synthesis rate, which they propose to be due to loose control of metabolism, which could be advantageous for short and long term adaptation of cells. This is an interesting study as our understanding of protein synthesis, metabolic and growth rate variation is very incomplete. Having said that, I disagree with some of the authors conclusions. For example, for me the finding that protein synthesis rate is highly variable is neither a surprising (as stated in the abstract) nor a new finding given the stochastic nature of metabolic reactions in single cells which has been observed already ten years ago (eg, Kiviet et al, 2014 and already cited in the manuscript). The strength of the manuscript is in the analysis of single cell variation in a well controlled model system and the single cell analyses based on the amino acid analog incorporation. On the other hand, the work is mostly descriptive with little mechanistic insights into the process. There is also quite a bit of speculative discussion, which I would like to have seen supported by more solid experimental evidence.

Major comments

First of all, I am not quite sure what the authors mean by the title "Regulation of variability in global cellular metabolism involving the mTOR pathway" as to me this title is ambiguous. Just reading the title, should it be "involves the mTOR pathway" or is the focus on that part of the metabolism directly involving mTOR in which case the word global is superfluous? The manuscript also focuses on protein synthesis as a readout rather than metabolism per se.

I am a little worried that the current title could be misleading.

How can it be explained that cell cycle regulators change the variability, while variability in the initial experiments was not affected by the cell cycle but still regulated in a cell cycle dependent manner? Is this independent of the cell cycle functions of these proteins? The authors speculate in the discussion that the cell cycle affects the TOR pathway through CDK(1?) but this needs more explanation and perhaps also some clear experimentation.

Related to above, what this manuscript is missing is a mechanistic understanding of the role of TOR pathway and cell cycle regulators. While not detrimental by itself, the cell cycle related results cause some confusion what is actually found related to the TOR pathway. Beyond the initial epistasis experiments, there is not much on this. This is particularly problematic as the genetic screen performed was a candidate screen containing many TOR pathway and cell cycle mutants so one would therefore expect to see these as hits given their overrepresentation in the screen and also due to the relaxed cut-off for identifying hits (see minor comments).

The authors speculate that since there is high variability in protein synthesis (as measured by AHA and HPG incorporation), there is no buffering against the impact of this. However, this conclusion is in my opinion premature. The study looked at total protein synthesis (or strictly speaking the amino acid analog transport and protein synthesis). Buffering against the impact of excessively low or high protein synthesis could be counteracted by protein degradation or differences in synthesis of specific proteins. Along these lines, the TOR and cell cycle pathways identified could simultaneously affect protein degradation thus maintaining the coordination of protein synthesis and degradation and having no effect on overall growth rate. For example, the TOR pathway is well known to be participating in both synthesis and degradation (both proteasome and lysosome).

Minor comments

Line 117 says "All strains were cultured in the same growth conditions and had similar growth rates.", but there is no actual evidence demonstrating this for all the strains. The EV3 suggests <10% differences, but how closely similar are the growth rates of all strains, not just those of selected examples?

For Fig 1E, F. As concluded, it looks like that the variability is not increased with cell length. This is an interesting although slightly unrelated point for this manuscript as this result suggests that protein synthesis variability may not be used to control cell size (eg, there is no cell size/mass specific metabolic rate to control cell size). Regardless, the CV as a function of cell length is not exactly a straight horizontal line suggesting there could be some effects. What is the error for the CV measurement?

Fig2. Given the importance of the homeostatic behavior at the population level, it would be necessary to swap the HPG and AHA labels for this experiment as an additional control experiment. Also, there could be alternative explanations for the observed lack of correlation. These are not discussed. Two potential explanations which immediately come to my mind are 1) contribution

by protein degradation to eliminate the first label incorporated into proteins and 2) regression of high protein biosynthesis to mean as the first label incorporation will generate dysfunctional proteins, which would reduce the fitness (and protein synthesis rate) of the cell when the second label is added.

Fig3. From a statistical perspective, the 1.5x SD of the mean WT QCD as the cut-off is quite relaxed leading to false positive findings although in the subsequent validation experiments (Fig4) the WT cells seem to be substantially more tightly distributed and resulting in significant differences between the strains. From where is the variability in QCD of the WT cells coming from, is it also stochastic? The large variability in the QCD of the WT cells reduces my confidence in the screen results as it could be possible that the QCDs can be substantially affected by some unknown environmental variables.

Referee #2:

Basier and Nurse investigate single-cell variability of the rate of protein synthesis. They demonstrate high variability in the short term, but homeostasis is quickly achieved on a population level. Furthermore, they address possible genetic determinants of the variability and show that cells can regulate the extent of the variability through the mTOR pathway and components of the cell-cycle control network.

The experimental approach is elegant, simple, yet powerful. The presentation of the data is very clear and easy to follow. There is nothing to fault on the experimental design or presentation of the results.

The questions they address are interesting and might have relevance for important issues such as drug resistance. However, the significance of the findings is less obvious. The authors refer to an accepted view that living organisms tend to have the ability to buffer the impact of random fluctuations and claim that their findings are in contradiction to this view. However, even if they show variability in protein synthesis rates in the short term, they also show that rates different from the population mean revert to the mean rate within 0.15 to 0.25 of the generation time. Therefore I fail to see the contradiction and the statement should be clarified or revised.

The stochastic nature of gene expression has been shown before both at the levels of transcription and translation. I believe the main novelty lies in the finding that the extent of the variability is regulated. The authors highlight the finding that the mTOR pathway is involved. However, the discussion of the data regarding the role of cell-cycle regulators is rather scanty. There is a lot of information in the analysis of the mutants, which is not discussed. In particular the results on CDK regulators should be discussed rather than just stated.

The Discussion contains a lot of interesting speculation regarding the benefits of the high variability, indeed I think it should be shortened or rather presented in another kind of article. The authors suggest that the high variability confers a selective advantage under changing environmental conditions. This is certainly a very attractive idea, although pure speculation at this stage. Having identified genetic determinants of the variability, the authors are in a position to experimentally address this idea, using the mutants displaying higher or lower variability. A correlation between the extent of variability and resilience in face of some stress or drug treatment would greatly support the hypothesis and strengthen the paper.

Minor comments

The description of the mutants is a bit short for readers not familiar with the fission yeast gene names.

Fig EV3F shows that *sck2Δ* has a much lower QCD than *psk1Δ* and *sck1Δ*, yet only the triple mutant is discussed in the text. Having identified which of the S6 kinases is involved in the regulation, why not point this out?

Referee #3:

The authors examine cell-to-cell variability in protein synthesis in a population of yeast cells. This and a series of follow-up experiments lead them to conclude that there is mTOR-controlled variability which constitutes a new organizational principle in complex metabolic systems. The experiments are clever and, to my understanding, well performed. The Discussion is balanced and particularly interesting. This reviewer is left wondering how the biology will be further pursued. It would be informative to investigate other metabolic pathways, particularly other mTOR-controlled pathways.

One minor point is the authors' use of the term 'mTOR'. mTOR usually refers to mammalian TOR. Many are now using mTOR in referring to the kinase from any organism, but this is not entirely correct or clear.

Referee #1:

The manuscript by Basier and Nurse identifies high variability in protein synthesis rate, which they propose to be due to loose control of metabolism, which could be advantageous for short and long term adaptation of cells. This is an interesting study as our understanding of protein synthesis, metabolic and growth rate variation is very incomplete. Having said that, I disagree with some of the authors conclusions. For example, for me the finding that protein synthesis rate is highly variable is neither a surprising (as stated in the abstract) nor a new finding given the stochastic nature of metabolic reactions in single cells which has been observed already ten years ago (eg, Kiviet et al, 2014 and already cited in the manuscript). The strength of the manuscript is in the analysis of single cell variation in a well controlled model system and the single cell analyses based on the amino acid analog incorporation. On the other hand, the work is mostly descriptive with little mechanistic insights into the process. There is also quite a bit of speculative discussion, which I would like to have seen supported by more solid experimental evidence.

We have replaced the word “surprising” as suggested by the Referee with “strikingly” in the abstract (line 15). We think that the high variability in the rate of protein synthesis is striking because it is a process dependent upon many hundreds of reactions, thus variability at the level of single genes would be expected to be averaged out at the cellular level. The majority of variability studies which have been described as stochastic have focussed more on relatively simple metabolic reactions with only a few steps, far fewer than for protein synthesis. Our work was with eukaryotic cells (Kiviet *et al.* worked on bacteria) and to our knowledge this is the first report on cellular protein

synthesis. In addition, variability in bacterial systems is not always conserved in eukaryotes, as has been shown for the ATP concentration for instance (PMID: 35961315 and PMID: 30858198). We are not sure we explained this well enough so have modified the text in several places to make this clearer. In the abstract (lines 15-16), the introduction (lines 57-63, 72-73), and the discussion (lines 241-246).

Our work describes a novel phenomenon, which is at too early a stage to come to mechanistic explanations (by which the Referee probably means a detailed “molecular” mechanism) but we have proposed a novel “conceptual” mechanism. We discuss conventional bet-hedging, but are not convinced that is the explanation. As an alternative we proposed the possibility that loose overall control might be needed to ensure agility and fluidity, representing a possible novel organisational principle. This mechanistic explanation is conceptual rather than molecular, and is a major point we have now emphasised more in the paper. Molecular mechanism is of course a long term aim as the Referee indicates and we have now discussed possible (although speculative) molecular explanations (lines 349-373, 381-387).

Major comments

First of all, I am not quite sure what the authors mean by the title "Regulation of variability in global cellular metabolism involving the mTOR pathway" as to me this title is ambiguous. Just reading the title, should it be "involves the mTOR pathway" or is the focus on that part of the metabolism directly involving mTOR in which case the word global is superfluous? The manuscript also focuses on protein synthesis as a readout rather than metabolism per se.

I am a little worried that the current title could be misleading.

We agree with the Referee that the title could be misleading and given their comment have changed it to “TOR-regulated variability in rate of cellular protein synthesis”.

How can it be explained that cell cycle regulators change the variability, while variability in the initial experiments was not affected by the cell cycle but still regulated in a cell cycle dependent manner? Is this independent of the cell cycle functions of these proteins? The authors speculate in the discussion that the cell cycle affects the TOR pathway through CDK(1?) but this needs more explanation and perhaps also some clear experimentation.

We agree with the Referee that it is not clear how cell cycle controllers could influence the variability, but we have now added to the manuscript a discussion of possible molecular mechanisms that could be involved. Relevant experiments we now refer to are that CDK1 can phosphorylate the translation initiation factor 4E-binding protein (4E-BP1) in human cells (PMC4434708) and that CDK1 regulator Cdc25 has been suggested to have a function linking growth rate and the cell cycle (Nurse & Thuriaux, 1984). Since the TOR pathway is a major controller of growth in cells, this Cdc25 function could be acting through the TOR pathway. There is also evidence that Pyp3 physically interacts and phosphorylates eIF4E (Tif1) in *S. pombe* (Yimit, 2011), also potentially linking modulation of variability to cell cycle controllers. We have added these possible explanations to the text (lines 349-373).

Related to above, what this manuscript is missing is a mechanistic understanding of the role of TOR pathway and cell cycle regulators. While not detrimental by itself, the cell

cycle related results cause some confusion what is actually found related to the TOR pathway. Beyond the initial epistasis experiments, there is not much on this. This is particularly problematic as the genetic screen performed was a candidate screen containing many TOR pathway and cell cycle mutants so one would therefore expect to see these as hits given their overrepresentation in the screen and also due to the relaxed cut-off for identifying hits (see minor comments).

Our screen was exploratory to determine if there were any genetic determinants of variability. Our experimental exploration involved analysis of nearly 50 gene deletions, each one of which is quite time consuming.

As discussed above, our work proposes a conceptual mechanism (it is not just descriptive), and it is too early to have a molecular mechanism in place for such a complex phenomenon that we described. Our epistasis experiment suggests that the TOR regulation of variability is downstream the cell cycle controllers' effect. However, as mentioned earlier we agree with the Referee that it would be useful to comment more on potential molecular mechanisms, and we have expanded the text to consider more how the TOR-related molecular mechanisms could be involved in the modulation of variability (lines 380-387).

With respect to the relaxed cut-off, this was used to analyse the imaging flow cytometer results to capture all possible potential regulators. But we then went on to perform multiple repeats of the more extreme hits using the more accurate fluorescence microscopy-based method. This more in-depth analysis confirmed that some of the strains we assayed did indeed have higher/lower variability and were not false positives, and so were worthy of discussion.

The authors speculate that since there is high variability in protein synthesis (as measured by AHA and HPG incorporation), there is no buffering against the impact of this. However, this conclusion is in my opinion premature. The study looked at total protein synthesis (or strictly speaking the amino acid analog transport and protein synthesis). Buffering against the impact of excessively low or high protein synthesis could be counteracted by protein degradation or differences in synthesis of specific proteins. Along these lines, the TOR and cell cycle pathways identified could simultaneously affect protein degradation thus maintaining the coordination of protein synthesis and degradation and having no effect on overall growth rate. For example, the TOR pathway is well known to be participating in both synthesis and degradation (both proteasome and lysosome).

We were discussing overall the measured rate of protein synthesis which would include protein degradation as the Referee suggests. To cover this aspect of net protein synthesis we now specifically mention protein turnover and degradation (lines 66-67, 244).

Minor comments

Line 117 says "All strains were cultured in the same growth conditions and had similar growth rates.", but there is no actual evidence demonstrating this for all the strains. The EV3 suggests <10% differences, but how closely similar are the growth rates of all strains, not just those of selected examples?

For all the strains investigated using fluorescence microscopy, we showed the growth rates had less than 10% deviation from the wild type. This is shown on Figure EV3C and D, and Figure EV4C and F. For the other strains used in the screen, all cultures were seeded at the same density and achieved at least at 80% of the growth compared with wild type after 17 hours of cultures (corresponding to approximately 4 rounds of division), also indicating a difference of less than 10% in growth rate. We now describe this in the manuscript (line 165).

For Fig 1E, F. As concluded, it looks like that the variability is not increased with cell length. This is an interesting although slightly unrelated point for this manuscript as this result suggests that protein synthesis variability may not be used to control cell size (eg, there is no cell size/mass specific metabolic rate to control cell size). Regardless, the CV as a function of cell length is not exactly a straight horizontal line suggesting there could be some effects. What is the error for the CV measurement?

Below is a calculation of the Standard Error of the Coefficient of Variation for Figure 1F. We do not think that we can conclude from this data that there is a significant deviation from a horizontal line.

Cell length (μm)	CV (%)	N	SE(CV)
[6,7[30.5	395	1.18
[7,8[31.5	1991	0.55
[8,9[31.9	2127	0.54

[9,10[32.0	1658	0.61
[10,11[32.5	1412	0.67
[11,12[30.0	1007	0.73
[12,13[29.4	502	1.00

Fig2. Given the importance of the homeostatic behavior at the population level, it would be necessary to swap the HPG and AHA labels for this experiment as an additional control experiment. Also, there could be alternative explanations for the observed lack of correlation. These are not discussed. Two potential explanations which immediately come to my mind are 1) contribution by protein degradation to eliminate the first label incorporated into proteins and 2) regression of high protein biosynthesis to mean as the first label incorporation will generate dysfunctional proteins, which would reduce the fitness (and protein synthesis rate) of the cell when the second label is added.

Swapping the labels would have been an additional experiment which we wanted to do, but unfortunately this turned out not to be possible because the chemical reaction between the HPG molecule and the fluorescent probe that clicks to it is much more efficient than the reaction between the AHA molecule and the fluorescent probe that binds to it. The first label is diluted too much within cells over time by growth. For this reason, the HPG analogue had to be the first label used. The experiment we did carry out exhibited strong correlation between HPG and AHA signals within single cells when co-pulsed (see Figure EV1F and Figure 2B at 0 min), so we are confident in the conclusions we have made.

With respect to the two potential explanations mentioned by the Referee, we found no evidence that 10 μ M led to significant toxicity in cells. It resulted in only about an 8 % overall difference in growth rate after 480 minutes (PMC10152140) which is much longer than the 120-minute time course used in this study so we think explanation (2) is unlikely. Proteins generally have long half-lives in fission yeast (>10 h, PMC4526151) so degradation is unlikely to have a major impact on the de-correlation given our time frame of 20-60 minutes.

Fig3. From a statistical perspective, the 1.5x SD of the mean WT QCD as the cut-off is quite relaxed leading to false positive findings although in the subsequent validation experiments (Fig4) the WT cells seem to be substantially more tightly distributed and resulting in significant differences between the strains. From where is the variability in QCD of the WT cells coming from, is it also stochastic? The large variability in the QCD of the WT cells reduces my confidence in the screen results as it could be possible that the QCDs can be substantially affected by some unknown environmental variables.

The Referee is quite correct that there is variability in the wild type cells in repeated experiments using the image flow cytometer. For that reason we only considered in-depth the more extreme outliers well beyond the span of the wild type strains, and we also repeated these outlier analyses using fluorescence wide field microscope to confirm the results.

Referee #2:

Basier and Nurse investigate single-cell variability of the rate of protein synthesis. They demonstrate high variability in the short term, but homeostasis is quickly achieved on a population level. Furthermore, they address possible genetic determinants of the variability and show that cells can regulate the extent of the variability through the mTOR pathway and components of the cell-cycle control network.

The experimental approach is elegant, simple, yet powerful. The presentation of the data is very clear and easy to follow. There is nothing to fault on the experimental design or presentation of the results.

The questions they address are interesting and might have relevance for important issues such as drug resistance. However, the significance of the findings is less obvious. The authors refer to an accepted view that living organisms tend to have the ability to buffer the impact of random fluctuations and claim that their findings are in contradiction to this view. However, even if they show variability in protein synthesis rates in the short term, they also show that rates different from the population mean revert to the mean rate within 0.15 to 0.25 of the generation time. Therefore I fail to see the contradiction and the statement should be clarified or revised.

We didn't wish to imply that living organisms cannot buffer the impact of random fluctuations and have therefore removed the sentence the Referee was concerned about: "This observation is in contrast with the view that living organisms tend to evolve mechanisms to buffer the impact of random fluctuations in their cellular processes". Concerning the significance of our findings and hedge betting, for example for drug resistance, we have now put greater emphasis in the Discussion on our

alternative explanation to hedge betting involving a proposed new conceptual mechanism concerning the regulation of complex metabolic cellular systems (lines 263-290).

The stochastic nature of gene expression has been shown before both at the levels of transcription and translation. I believe the main novelty lies in the finding that the extent of the variability is regulated. The authors highlight the finding that the mTOR pathway is involved. However, the discussion of the data regarding the role of cell-cycle regulators is rather scanty. There is a lot of information in the analysis of the mutants, which is not discussed. In particular the results on CDK regulators should be discussed rather than just stated.

As the Referee says, the stochastic nature of gene expression has been shown often before for transcription and translation, but this has primarily been at the relatively simple single mRNA/protein level. Our work has shown variability of global protein synthesis involving many hundreds of reactions. It is expected that at the scale of the whole cell, large numbers of individual high variabilities acting together will in part 'cancel' each other, resulting in lower variability. We show is not the case and that there is high variability overall.

It is not clear how cell cycle controllers could influence the variability. Relevant experiments are that CDK1 can phosphorylate the translation initiation factor 4E-binding protein (4E-BP1) in human cells (PMC4434708) and that CDK1 regulator Cdc25 has been suggested to have a function linking growth rate and the cell cycle (Nurse & Thuriaux, 1984). Since the TOR pathway is a major controller of growth in cells, this Cdc25 function could be acting through the TOR pathway. There is also evidence that

Pyp3 physically interacts and phosphorylates eIF4E (Tif1) in *S. pombe* (Yimit, 2011), also potentially linking modulation of variability to cell cycle controllers. We have modified the manuscript to expand on these points and now discuss the role of the cell cycle regulators more fully (lines 15-16, 57-63, 72-73, 241-246, 351-373).

The Discussion contains a lot of interesting speculation regarding the benefits of the high variability, indeed I think it should be shortened or rather presented in another kind of article. The authors suggest that the high variability confers a selective advantage under changing environmental conditions. This is certainly a very attractive idea, although pure speculation at this stage. Having identified genetic determinants of the variability, the authors are in a position to experimentally address this idea, using the mutants displaying higher or lower variability. A correlation between the extent of variability and resilience in face of some stress or drug treatment would greatly support the hypothesis and strengthen the paper.

It will be interesting to test correlations between the extent of variability and resilience in conditions of stress. However, finding the right kind of stress for this experiment will likely to require extensive new work beyond the scope of the present paper.

For us, an important aspect of the present paper is that variability may go beyond well discussed bet-hedging explanations and could reveal a novel organisational principle that is 'loose' regulation to ensure agility. Because this is novel it forms an essential part of the discussion. This is made clearer in the revised version (lines 263-290).

Minor comments

The description of the mutants is a bit short for readers not familiar with the fission yeast gene names.

We agree this was short and refer readers to specific references if they require more information.

Fig EV3F shows that *sck2Δ* has a much lower QCD than *psk1Δ* and *sck1Δ*, yet only the triple mutant is discussed in the text. Having identified which of the S6 kinases is involved in the regulation, why not point this out?

We have changed the text to point this out (lines 196-197).

Referee #3:

The authors examine cell-to-cell variability in protein synthesis in a population of yeast cells. This and a series of follow-up experiments lead them to conclude that there is mTOR-controlled variability which constitutes a new organizational principle in complex metabolic systems. The experiments are clever and, to my understanding, well performed. The Discussion is balanced and particularly interesting. This reviewer is left wondering how the biology will be further pursued. It would be informative to investigate other metabolic pathways, particularly other mTOR-controlled pathways.

One minor point is the authors' use of the term 'mTOR'. mTOR usually refers to mammalian TOR. Many are now using mTOR in referring to the kinase from any organism, but this is not entirely correct or clear.

We have changed 'mTOR' to 'TOR' as the Referee suggests. We have also expanded the discussion about the molecular mechanisms and metabolic pathways that could be involved (lines 351-373, 381-387).

Dear Dr Clovis Basier, dear Dr Paul Nurse,

Thank you for submitting your revised manuscript (EMBOJ-2023-115967R) to The EMBO Journal. Your amended study was sent back to the three referees for their scientific re-evaluation, and we have received comments from all of them, which I enclose below.

As you will see, the experts state that the work has been substantially improved by the revisions and they are now broadly in favour of publication.

Thus, we are pleased to inform you that your manuscript has been accepted in principle for publication in The EMBO Journal.

We now need you to take care of a number of issues related to formatting and data presentation as detailed below, which should be addressed at re-submission.

Please contact me at any time if you have additional questions related to below points.

As you might have seen on our web page, every paper at the EMBO Journal now includes a 'Synopsis', displayed on the html and freely accessible to all readers. The synopsis includes a 'model' figure as well as 2-5 one-short-sentence bullet points that summarize the article. I would appreciate if you could provide this figure and the bullet points.

Thank you for giving us the chance to consider your manuscript for The EMBO Journal. I look forward to your final revision.

Again, please contact me at any time if you need any help or have further questions.

Best regards,

Daniel Klimmeck

>> Please add up to five keywords for your study.

>> Author Contributions: Please remove the author contributions information from the manuscript text. Note that CRediT has replaced the traditional author contributions section as of now because it offers a systematic machine-readable author contributions format that allows for more effective research assessment. and use the free text boxes beneath each contributing author's name to add specific details on the author's contribution.

More information is available in our guide to authors.
<https://www.embopress.org/page/journal/14602075/authorguide>

>>Manuscript composition: the order of the manuscript sections should be: Abstract / Keywords / Introduction / Results / Discussion / Materials and Methods / Data Availability / Acknowledgements / Disclosure and competing interests statement / References / Figure legends / Expanded View Figure legends.

>> Figures: The figures should be removed from the manuscript.

>> Please consistently indicate the funding information in our online system. Currently missing: 'the Lord Leonard and Lady Estelle Wolfson Foundation'.

>> Dataset EV Legends: Table EV1 should be removed from the manuscript and uploaded as an individual expanded view file.

>> Please add a completed Author Checklist to your manuscript.

>> Revisit publication status of the bioRxiv reference Liu et al (2021) and update in case of formal journal publication.

>> Consider additional changes and comments from our production team as indicated below:

- Figure legends:

Please indicate the statistical test used for data analysis in the legends of figure 2b; EV 1e-f.

Referee #1:

I am happy with the manuscript after reading the additional explanations provided by the authors. I agree that the manuscript contains several important concepts and these, and that this is also the first paper on variation in protein synthesis in eukaryotes, are all important additions to the literature and will likely have positive impact in the thinking of the wider scientific community.

Thank you for also pointing out that there is a technical limitation if one would wish to perform the label-swap, such information is useful and might be worth mentioning in the methods to help the scientific community as these labeling kits are also relatively expensive.

I have no further comments.

Referee #2:

The authors have addressed all my concerns. The clearer highlighting of the novel aspects, as well as the extended discussion regarding the possible molecular mechanisms involved have greatly improved the manuscript.

Referee #3:

The authors have presented a novel concept concerning a new organizational principle in metabolism. Whereas it is clear that more molecular explanation of this new concept is needed, it is also reasonable to accept that the concept is important and worthy of publication as is. This author is of the latter camp. This study plants a seed, i.e, it could be seminal. It does not intend to provide a molecular explanation.

The authors addressed the remaining editorial issues.

Dear Dr Basier, dear Dr Nurse,

Thank you for submitting the revised version of your manuscript. I have now evaluated your amended manuscript and concluded that the remaining minor concerns have been sufficiently addressed.

I am pleased to inform you that your manuscript has been accepted for publication in the EMBO Journal.

On a different note, I would like to alert you that EMBO Press offers a format for a video-synopsis of work published with us, which essentially is a short, author-generated film explaining the core findings in hand drawings, and, as we believe, can be very useful to increase visibility of the work. Please see the following link for representative examples and their integration into the article web page:

<https://www.embopress.org/doi/full/10.15252/emj.2019103932>

If you have any questions, please do not hesitate to contact the Editorial Office.

Best regards,

Daniel Klimmeck

Daniel Klimmeck, PhD
Senior Editor
The EMBO Journal
EMBO
Postfach 1022-40
Meyerohofstrasse 1
D-69117 Heidelberg
contact@embojournal.org
Submit at: <http://emboj.msubmit.net>
